# Efficient Bisection Projection to Ensure Neural-Network Solution Feasibility for Optimization over General Set

**Enming Liang** [1]    **Minghua Chen** [1][2]

## Abstract

Neural networks (NNs) have emerged as promising tools for solving constrained optimization problems in real-time. However, ensuring constraint satisfaction for NN-generated solutions remains challenging due to prediction errors. Existing methods to ensure NN feasibility either suffer from high computational complexity or are limited to specific constraint types. We present Bisection Projection, an efficient approach to ensure NN solution feasibility for optimization over general compact sets with non-empty interiors. Our method comprises two key components: (i) a dedicated NN (called IPNN) that predicts interior points (IPs) with low eccentricity, which naturally accounts for approximation errors; (ii) a bisection algorithm that leverages these IPs to recover solution feasibility when initial NN solutions violate constraints. We establish theoretical guarantees by providing sufficient conditions for IPNN feasibility and proving bounded optimality loss of the bisection operation under IP predictions. Extensive evaluations on real-world non-convex problems demonstrate that Bisection Projection achieves superior feasibility and computational efficiency compared to existing methods, while maintaining comparable optimality gaps.

## 1. Introduction

Constrained Optimization (CO) plays an essential role in various engineering fields, such as supply chain, transportation, and power systems. To solve CO problems, iterative algorithms, such as interior point methods, have been developed and embedded within commercial solvers like Gurobi. These tools are designed to tackle CO with high precision, providing high-quality solutions. However, they can be slow for real-time applications with tight time constraints.

Recent advancements in machine learning (ML) have introduced innovative strategies for solving CO problems in real-time, including the end-to-end (E2E) solution mapping (Amo22), the learning-to-optimize (L2O) iterative scheme (CCC+21), the generative modeling framework (LC23), and hybrid approaches (KFVHW21). One powerful idea is to leverage the universal approximation ability of neural networks (NNs) (HSW89; LLPS93) to predict high-quality solutions given input parameters, significantly reducing computation time compared to iterative solvers. For instance, NNs have been trained to solve non-convex optimal power flow problems in grid operations, achieving 2-4 orders of magnitude speedup over iterative solvers (PZC19; GWWM19; PZCZ20; FMVH20; ZB20; DRK20).

Despite the minor optimality loss and significant speedup of NN-based methods, guaranteeing the feasibility of NN solutions with respect to problem constraints, which is important to safe-critical scenarios, remains a challenge due to inherent NN prediction errors. Existing methods to ensure NN feasibility either suffer from high computational complexity or are limited to specific constraints. See Sec . 2 for discussions on related works.

In this paper, we develop **Bisection Projection** (BP) as a simple yet efficient scheme to recover NN solutions feasibility over a (fairly) general compact set with a non-empty interior, beyond previous works on linear (WZG+23; ZPCL23), convex (THH23), or ball-homeomorphic constraints (LCL23; LCL24). Our contributions are as follows:

▷ In Sec. 4, we introduce the BP framework for ensuring NN solution feasibility. It comprises two steps: (i) a dedicated NN (called IPNN) that predicts interior points (IPs) of low *eccentricity*; (ii) an efficient *bisection* algorithm that leverages these IPs to recover solution feasibility.

▷ In Sec. 5, we establish theoretical guarantees by providing sufficient conditions for IPNN feasibility and proving bounded *eccentricity*-related optimality loss of the bisection operation under IP predictions.

▷ In Sec. 6, we carry out extensive experiments over convex and non-convex problems to evaluate the performance of BP. The results show that it outperforms existing methods in feasibility and run-time complexity while achieving similar optimality losses.

[1]Department of Data Science, City University of Hong Kong, Hong Kong [2]School of Data Science, The Chinese University of Hong Kong (Shenzhen), China. Correspondence to: Minghua Chen <minghua.chen@cityu.edu.hk, minghua@cuhk.edu.cn>.

*Proceedings of the 42nd International Conference on Machine Learning*, Vancouver, Canada. PMLR 267, 2025. Copyright 2025 by the author(s).

*Table 1.* Existing work for ensuring NN solution feasibility for continuous constrained optimization problems.

| Existing Work (refer to Sec. 2 for details) | Constraint Setting | | | Performance Guarantee | | |
|---|---|---|---|---|---|---|
| | Input dependent | Non-linear equality | Non-convex inequality | Feasibility ensuring | Optimality bound | Low run-time |
| Penalty training | ✓ | ✓ | ✓ | ✗ | ✗ | ✓ |
| Orthogonal Proj. | ✓ | ✓ | ✓ | ✓ | ✓ | ✗ |
| Preventive learning | ✓ | ✗ (linear) | ✗ (linear) | ✓ | ✗ | ✓ |
| Gauge mapping | ✓ | ✗ (linear) | ✗ (linear) | ✓ | ✗ | ✓ |
| DC3 | ✓ | ✓ | ✓ | ✗ | ✗ | ✓ |
| Gauge Proj. | ✗ | ✗ (linear) | ✗ (convex) | ✓ | ✓ | ✓ |
| Homeomorphic Proj. | ✓ | ✓ | ✓ (BH[1]) | ✓ | ✓ | ✓ |
| **Bisection Proj.** | ✓ | ✓ | ✓ | ✓ | ✓ | ✓ |

[1] BH indicates the constraint set is homeomorphic to a ball, which includes all compact convex sets and a part of non-convex sets.

## 2. Related Work

Machine Learning (ML)-driven optimization has emerged as an active research field (PZC19; KFVHW21; CCC+21; Amo22). A fundamental challenge is ensuring neural network (NN) solution feasibility under input-dependent constraints. For basic constraints like simplexes or boxes, feasibility can be enforced through activation functions (e.g., Softmax or Sigmoid). For complex constraints, various approaches have been developed, as summarized in Table 1.

**Equality constraints**. Linear equations and certain non-linear equations with constant ranks, can be embedded as neural network layers by predicting a subset of variables and solving for the remaining variables to satisfy the equality constraints (Aba69; PZC19; DRK20; LCL23; DWDS23).

**Warm-start approach**. The NN predictions can serve as warm-start points for iterative solvers, potentially reducing the number of iterations required to reach the optimal/feasible solution (Die19; Bak19; SHAS24; GXW+24).

**Penalty training**. To reduce constraint violations of predicted solutions, various penalty functions, such as quadratic penalties, have been incorporated into the NN loss function (COMB19; PZCZ20; ZB20; FMVH20). Additionally, integration of the Karush–Kuhn–Tucker (KKT) conditions as equality constraints has been explored to refine NN performance (NC21b; NC21a; ZCZ21). However, these methods do not ensure solution feasibility.

**Provable training/verification**. A Preventive Learning framework is proposed for ensuring linear constraint feasibility without post-processing in (ZPC+20; ZPCL23), which adjusts inequality constraints to account for NN prediction errors. A NN editing method applies parametric linear relaxation to find NN weights and ensure output feasibility over the polytope (TT24). Additionally, NN verification techniques can also be applied to assess the worst-case performance after training (VQLC20; uAYKJ22; LAL+21).

**Set representation/approximation**. To guarantee feasibility, an inner approximation of the original constraint set can be constructed. The convex combination of vertices and rays can represent feasible points of linear constraints (FNC20; ZSRZ21). For general compact constraint sets, the *probabilistic transformer* utilizes collected feasible samples to ensure feasibility (KZLD21). However, scalability remains a challenge due to the exponential growth in required samples with increasing dimensionality.

**Projection approach**. To enforce solution feasibility, orthogonal/L2 projection is often employed. However, solving projection problems either by iterative solvers or equivalent *optimization layers* (AK17; AAB+19; CDB+21; WZG+23; ZYZ+24) over complex constraints is computationally intensive. Alternative strategies include gradient-based methods (e.g., DC3 (DRK20)) and L2O models (HWFGY21; HFL+22). However, those projection-analogous methods do not guarantee feasibility for general constraints.

**Homeomorphic Projection** ensures NN solution feasibility over *ball-homeomorphic* (BH) constraints by constructing a homeomorphism between the constraint set and a unit ball using invertible NN, allowing efficient projection via bisection (LCL23; LCL24).

**Gauge function**. These works utilize gauge/Minkowski functions (BM08) to conduct projection or constrain NN output. For fixed convex sets, given an interior point, *gauge projection* can be applied to find feasible boundary solutions (Mha22). Similar approaches have been applied in the following works (LBGH23; THH23; KU23; LM23; TVH24). The gauge function can also be used to construct a bijection, known as gauge mapping, to constrain the NN output within a polytope (TZ22a; TZ22b; LKM23; LLC25).

In summary, existing approaches either have limited applicability or incur significant computational overhead. We propose **Bisection Projection** as a simple yet efficient framework to ensure NN solution feasibility over (fairly) general compact sets while maintaining bounded optimality loss. While algorithmically related to gauge-based methods, our method offers broader applicability and provides rigorous theoretical analysis.

## 3. Setting and Open Issue

We consider the following *continuous* optimization problem

$$\min_{x\in\mathbb{R}^n}\ f(x,\theta)\quad\text{s.t.}\ \ x\in\mathcal{C}_\theta,\qquad(1)$$

where $x\in\mathcal{C}_\theta\subset\mathbb{R}^n$ denotes the decision variable and $\theta\in\Theta\subset\mathbb{R}^d$ represents the input contextual parameter. Without loss of generality, the input domain $\Theta$ and constraint set $\mathcal{C}_\theta$ are assumed to be *compact*. The objective function $f(x,\theta)$ is *Lipschitz* continuous and potentially *non-convex*, with optimal solution denoted as $x_\theta^*\in\arg\min_{x\in\mathcal{C}_\theta}\{f(x,\theta)\}$. The constraint set $\mathcal{C}_\theta$ is specified by a set of inequalities: $g(x,\theta)=[g_1(x,\theta),\dots,g_{n_{\text{ineq}}}(x,\theta)]\le 0$.

While equality constraints are not explicitly included in this formulation, certain classes of equality constraints of a constant rank can be embedded as neural network layers and satisfied exactly (Aba69; Lee13; PZC19; DRK20; LCL23; DWDS23). We do carry out simulations for problems with linear/nonlinear equality constraints in Sec. 6 and provide detailed discussion in Appendix A.4.

We further specify the constraint set as follows, beyond those discussed in related works in Sec. 2.

**Assumption 1.** $\forall\theta\in\Theta$, (i) the constraint set $\mathcal{C}_\theta$ has positive measure [1]; (ii) the set-valued mapping $\theta\mapsto\mathcal{C}_\theta$ is continuous in the Hausdorff metric [2].

We remark that assumption (i) guarantees the existence of interior points, which is fundamental to the bisection approach detailed in Sec. 4.1; and assumption (ii) ensures continuous deformation of the constraint set with respect to the input parameter, enabling the learning of continuous mappings from inputs to interior points as discussed in Sec. 4.3. These assumptions deliberately exclude unusual sets (e.g., vertices of a hypercube) from consideration in our *continuous* optimization problem, enabling rigorous algorithmic development and theoretical analysis. Despite these restrictions, Assumption 1 encompasses a broad class of constraint sets, including linear, convex, and ball-homeomorphic sets studied in existing works (ZPCL23; WZG$^+$23; LCL23; THH23; LCL24).

**Open issue.** As discussed in Sec. 2, various NN-based methods have demonstrated success in solving constrained optimization problems, offering low run-time complexity and small optimality gaps (PZC19; DRK20; PVH23; HCL24). We denote one such trained NN predictor as $F(\theta)$ : $\mathbb{R}^d\ \to\ \mathbb{R}^n$ and define its prediction error as $\epsilon_{\text{pre}}=\sup_{\theta\in\Theta}\{\|F(\theta)-x_\theta^*\|\}$. Due to the error $\epsilon_{\text{pre}}$, ensuring NN solution feasibility is non-trivial. As illustrated in Fig.

---

[1] i.e., $\forall x\in\text{int}(\mathcal{C}_\theta),\exists\epsilon>0$ such that $\mathcal{B}(x,\epsilon)\subseteq\mathcal{C}_\theta$, where $\mathcal{B}(x,\epsilon)$ is the open ball centered at $x$ with radius $\epsilon$.

[2] i.e., $\forall\theta_0\in\Theta,\forall\epsilon>0,\exists\delta>0$ such that $d_H(\mathcal{C}_\theta,\mathcal{C}_{\theta_0})<\epsilon$ whenever $\|\theta-\theta_0\|<\delta$, where $d_H$ denotes the Hausdorff distance.

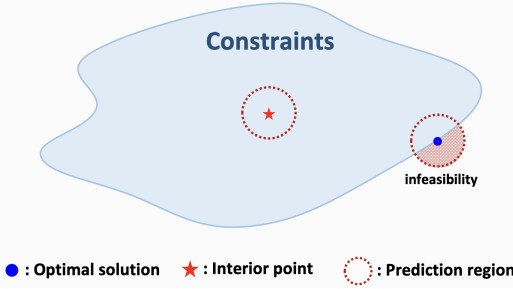

*Figure 1.* NN predicting optimal solution (on boundary) may incur infeasibility; NN predicting interior points accommodates errors.

1, an optimal solution often lies on the constraint boundary with active constraints, making NN feasibility particularly challenging — *Any positive prediction errors can push NN predictions outside of feasible regions.*

Current approaches, as summarized in Table 1, are either computationally intensive or fail to provide performance guarantees over general input-dependent constraint sets. To date, ensuring NN solutions feasibility for constraint optimization in (1) under Assumption 1, while maintaining bounded optimality loss and low computational complexity, remains an open and pressing challenge.

## 4. The Bisection Projection Framework

We propose *Bisection Projection* (BP) to "project" infeasible NN solutions onto the constraint set with low run-time complexity and minor optimality loss. As detailed in Sec. 4.1, this framework applies bisection to iteratively narrow the gap between infeasible points and one interior point (IP) to find feasible solutions. In Sec. 4.2, to bound the optimality loss induced by bisection, we introduce the concept of *eccentricity* for IP and establish its connection to projection distance. In Sec. 4.3, to reduce the inference time for finding IPs under varying inputs, we train another NN, denoted as IPNN, to predict IPs in real time.

### 4.1. Bisection with Interior Points

Given an infeasible NN prediction $\tilde{x}_\theta\notin\mathcal{C}_\theta$ and an IP $x_\theta^\circ\in\mathcal{C}_\theta$, we can "project" $\tilde{x}_\theta$ to $\mathcal{C}_\theta$ as:

$$\hat{x}_\theta=\text{BP}(\tilde{x}_\theta,x_\theta^\circ)\triangleq\alpha^*\cdot(\tilde{x}_\theta-x_\theta^\circ)+x_\theta^\circ,\qquad(2)$$

where $\alpha^*\in[0,1]$ leads to $\hat{x}_\theta\in\partial\mathcal{C}_\theta$ and $\partial\mathcal{C}_\theta$ is the boundary of $\mathcal{C}_\theta$. As illustrated in Fig. 2, the "projected" solution $\hat{x}_\theta$ is located on the straight line segment connecting the infeasible solution $\tilde{x}_\theta$ and an IP $x_\theta^\circ$. We note that there could be multiple $\alpha^*$ and corresponding $\hat{x}_\theta$, given a pair of $\tilde{x}_\theta$ and $x_\theta^\circ$. To determine one such $\alpha^*$, we employ the bisection method, as elaborated in Alg. 1. We initiate by drawing a straight line connecting $\tilde{x}_\theta$ with an IP $x_\theta^\circ$. This segment is guaranteed to intersect the boundary of the feasible region

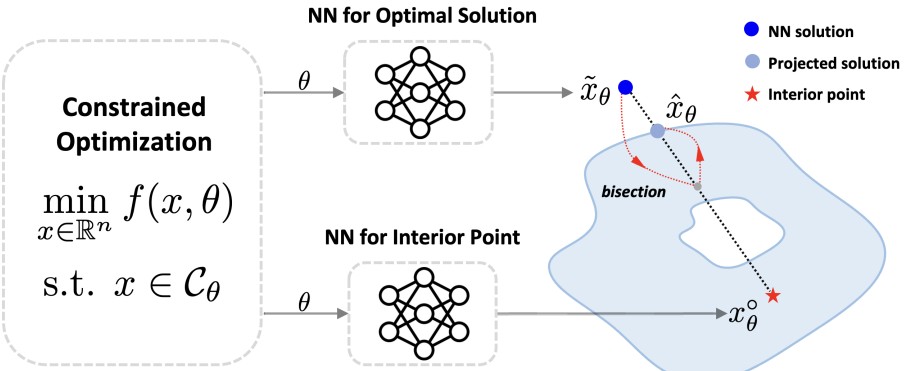

*Figure 2.* The **Bisection Projection** framework: we apply one NN to predict a solution for the constraint optimization problem (near-optimal but may not be feasible), and another NN to predict interior points (robust and feasible); then we apply bisection (Alg. 1) to recover solution feasibility.

at least once. Subsequently, we apply the bisection algorithm to iteratively pinpoint one feasible solution along this segment toward the constraint boundary. Such a bisection operation is applicable to general compact sets with non-empty interiors (Assumption 1), and is also efficient with a linear convergence rate and maintains low per-iteration computational cost involving feasibility checks.

*Remark* 1. We note that the concept of projection-analogous operation with IPs is non-sophisticated, and some works also leverage similar ideas to "project" infeasible prediction (Mha22; THH23; KU23; LM23; TVH24; LCL24), see Appendix A.1 for details of these methods. However, these works primarily focus on fixed convex constraint sets where a single IP can be computed offline for online deployment. Two critical gaps remain in the literature: (i) theoretical analysis of how IP selection influences the quality of projected solutions; (ii) developing efficient schemes to obtain IPs for general input-dependent constraint sets in real-time.

Our work advances these fundamental issues by: (i) we first introduce the eccentricity of IP and establish its connection to the bisection-induced projection distance in Sec. 4.2. (ii) We then employ another NN, called IPNN, to efficiently predict IPs under varying inputs for input-dependent constraint in Sec. 4.3. We also present a comprehensive performance analysis in Sec. 5.

### 4.2. Minimum-Eccentricity IP for Bisection

We first define the eccentricity of IP, which is crucial for bounding the bisection-induced projection distance.

**Definition 4.1** (Eccentricity of IP). For a compact set $\mathcal{C}_\theta$ satisfying Assumption 1 with a non-empty interior, the eccentricity for one IP $x_\theta^\circ \in \mathcal{C}_\theta$ with respect to a compact subset of boundary $\Gamma \subseteq \partial \mathcal{C}_\theta$ is defined as:

$$\mathcal{E}(x_\theta^\circ, \Gamma) \triangleq \max_{y \in \Gamma} \|y - x_\theta^\circ\| - \min_{y \in \Gamma} \|y - x_\theta^\circ\|. \quad (3)$$

---

**Algorithm 1** Bisection for Feasibility.

**Input:** an NN prediction $\tilde{x}_\theta \notin \mathcal{C}_\theta$ and an IP $x_\theta^\circ \in \mathcal{C}_\theta$

1: set total iteration $K$, $\alpha_l = 0$, and $\alpha_u = 1$
2: **for** $n = 1 : K$ **do**
3:     bisection: $\alpha_m = (\alpha_l + \alpha_u)/2$
4:     **if** $x_\theta^\circ + \alpha_m \cdot (\tilde{x}_\theta - x_\theta^\circ) \in \mathcal{C}_\theta$ **then**
5:         increase lower bound: $\alpha_l \leftarrow \alpha_m$
6:     **else**
7:         decrease upper bound: $\alpha_u \leftarrow \alpha_m$
8:     **end if**
9: **end for**

**Output:** feasible solution $\hat{x}_\theta = \alpha_l \cdot (\tilde{x}_\theta - x_\theta^\circ) + x_\theta^\circ \in \mathcal{C}_\theta$

---

When $\Gamma = \partial \mathcal{C}_\theta$, eccentricity quantifies the variation in point-to-boundary distances, effectively measuring the interior point's centrality within the feasible region. For instance, the center of a unit ball achieves zero eccentricity. This measure directly relates to the optimality loss in bisection projection — lower eccentricity corresponds to smaller bisection-induced optimality gaps.

When $\Gamma \subset \partial \mathcal{C}_\theta$ is a subset of the boundary, eccentricity provides a localized characterization focused on specific boundary regions of interest. Particularly, when $\Gamma$ encompasses the projected solutions of neural network predictions with bounded errors, this local eccentricity measure yields tighter bounds on the optimality loss compared to the global eccentricity measure (i.e., $\mathcal{E}(x_\theta^\circ, \Gamma_1) \leq \mathcal{E}(x_\theta^\circ, \Gamma_2)$ if $\Gamma_1 \subseteq \Gamma_2$).

Next, we establish the connection between eccentricity measure and the bisection-induced projection distance.

**Proposition 4.1.** *Let $\tilde{x}_\theta = F(\theta)$ be an infeasible NN prediction with bounded prediction error as $\|F(\theta) - x_\theta^*\| \leq \epsilon_{\text{pre}}$; $\hat{x}_\theta = \text{BP}(\tilde{x}_\theta, x_\theta^\circ)$ be the projected solution with IP $x_\theta^\circ \in \mathcal{C}_\theta$; Then, the worst-case projection distance is bounded as:*

$$\max_{\tilde{x}_\theta \in \mathcal{B}(x_\theta^*, \epsilon_{\text{pre}})} \|\tilde{x}_\theta - \text{BP}(\tilde{x}_\theta, x_\theta^\circ)\| \leq \epsilon_{\text{pre}} + \mathcal{E}(x_\theta^\circ, \Gamma_\theta),$$

where $\mathcal{B}(x_\theta^*, \epsilon_{\text{pre}})$ *represents the NN prediction region, containing all possible NN predictions with an error* $\epsilon_{\text{pre}}$*; and* $\Gamma_\theta = \{\text{BP}(\tilde{x}_\theta, x_\theta^\circ), \forall \tilde{x}_\theta \in \mathcal{B}(x_\theta^*, \epsilon_{\text{pre}}) \setminus \mathcal{C}_\theta\}$ *defines a subset of the constraint boundary, containing all "projected" infeasible NN predictions.*

The complete proof and a geometric illustration are included in Appendix B.1. We remark that the applicability of this bound extends to general compact sets under Assumption 1. Further, informed by Prop. 4.1, we seek to find an IP with minimized eccentricity (MEIP) with respect to constraint boundary $\partial\mathcal{C}_\theta$ to reduce the *worst-case* [3] bisection-induced projection distance for any infeasible NN solutions.

However, computing MEIP exactly presents significant challenges even for convex constraints, due to the non-convex boundary constraints in (26). To address this, we can bound the eccentricity by finding a surrogate central point[4] such as the *Chebyshev center* (BBV04), defined as the IP maximizing the minimum distance to the boundary: $\max_{x_\theta^\circ \in \mathcal{C}_\theta} \{\min_{y \in \partial\mathcal{C}_\theta} \|y - x_\theta^\circ\|\}$, which is also equivalent to minimizing the second term of the eccentricity in Def. 4.1.

Thus, this Chebyshev center serves as a *relaxation* for MEIP and provides an upper bound on the eccentricity. Moreover, it admits a tractable reformulation as the center of the largest Euclidean ball that can be inscribed in $\mathcal{C}_\theta$, which eliminates the non-convex boundary constraints in (26):

$$\max_{x_\theta^\circ \in \mathcal{C}_\theta} \gamma, \quad \text{s.t. } \mathcal{B}(x_\theta^\circ, \gamma) \subseteq \mathcal{C}_\theta \qquad (4)$$

However, computing the Chebyshev center for constraint set $\mathcal{C}_\theta$ under varying input $\theta$ poses significant computational challenges, particularly when rapid response times are essential for online applications. To overcome this limitation, we develop a learning-based approach in the following section. We train another neural network (denoted as IPNN) offline to learn the mapping from input parameters to the Chebyshev centers, thereby reducing the computational burden of finding interior points during real-time deployment.

### 4.3. Interior Points Neural Network (IPNN) Training

We utilize another neural network, denoted as IPNN $\psi(\cdot)$ : $\mathbb{R}^d \to \mathbb{R}^n$, to predict the IPs. We remark that such a continuous mapping from input to some IPs of interest exists under Hausdorff continuity specified in the Assumption 1.

---

[3]We consider minimize eccentricity $\mathcal{E}(x_\theta^\circ, \partial\mathcal{C}_\theta)$ with respect to the entire constraint boundary, which is *prediction-agnostic* and provides robust performance across different NN predictors over the same constraint set. In the appendix C, we also discuss *prediction-aware* eccentricity minimization when NN predictions or optimal solution data are given in advance.

[4]We provide a comprehensive review of different center definitions in Appendix A.3 to justify the design of MEIP.

**Algorithm 2** IPNN Training.

**Input:** Input data $\{\theta_i\}_{i=1}^N \in \Theta$, unit ball samples $\{u_i\}_{i=1}^M$
 1: Training epoch $E$, batch size $B$, IPNN $\psi$
 2: Initialize robust margins $\gamma = 10^{-2}$
 3: **for** $e = 1 : E$ **do**
 4: $\quad$ Sampling batched data: $\{\theta_i\}_{i=1}^B, \{u_i\}_{i=1}^B$
 5: $\quad$ Loss: $\mathcal{L}(\psi, \gamma) = \frac{1}{B}\sum_{i=1}^B \text{P}(x_{\theta_i}^\circ + \gamma \cdot u_i, \theta_i) - \lambda \cdot \log(\gamma)$
 6: $\quad$ Model update: $(\psi, \gamma) \leftarrow \text{Adam}(\mathcal{L}(\psi, \gamma))$
 7: **end for**
**Output:** Trained IPNN $\psi$.

We design the following loss function for IPNN training:

$$\mathcal{L}(\psi(\theta), \gamma) = \mathbb{E}_u[\text{P}(x_\theta^\circ + \gamma \cdot u, \theta)] - \lambda \cdot \log(\gamma) \qquad (5)$$

where the IP prediction is denoted as $x_\theta^\circ = \psi(\theta)$. The first loss term, $\text{P}(x_\theta^\circ + \gamma \cdot u, \theta) = \|g(x_\theta^\circ + \gamma \cdot u, \theta)^+\|$, denotes the constraint violation under the perturbed prediction with random samples $u$ from a unit ball, and its expectation represents the penalty for the inscribed ball constraint $\mathcal{B}(x_\theta^\circ, \gamma) \subseteq \mathcal{C}_\theta$ in (4). The regularization term, $\log(\gamma)$, represents the radius maximization objective in (4) and is adjusted by a positive coefficient $\lambda$.

We also note that the loss function is analogous to the adversarial learning techniques like *randomized smoothing* (CRK19) with Gaussian perturbations and fixed $\gamma$, ensuring the predicted IPs maintain a robust margin $\gamma$ from the constraint boundary. Finally, to optimize the average performance across different input parameters, we uniformly sample $\theta \in \Theta$ and minimize the total loss as $\mathcal{L}(\psi, \gamma) = \mathbb{E}_\theta[\mathcal{L}(\psi(\theta), \gamma)]$. The IPNN training procedure outlined in Alg. 2 involves sampling input parameters $\theta$ and unit vectors $u$ and follows regular NN training procedures.

## 5. Performance Analysis

In this section, we present a comprehensive analysis of the BP framework: (i) the sufficient conditions for IPNN training for producing feasible IP under any input parameter in Sec. 5.1; (ii) the optimality loss and run-time complexity for the bisection operation in Sec. 5.2. We also discuss the connection to existing approaches (TZ22b; LCL23; THH23) and IPNN training guarantees in Sec. 5.3.

### 5.1. Sufficient Conditions for IPNN Feasibility

**Proposition 5.1.** *Let $\mathcal{D}$ be an $r_\theta$-covering dataset for input domain as $\Theta \subseteq \bigcup_{i=1}^N \mathcal{B}(\theta_i, r_\theta)$; the constraint violation function $G(x, \theta) = \max_{1 \le j \le n_{ineq}} \{g_j(x, \theta)\}$ with Lipschitz $L_{G,x}$ and $L_{G,\theta}$ for $x$ and $\theta$, respectively; and IPNN $\psi$ is $L_\psi$-Lipschitz continuity for $\theta$. if $G(\psi(\theta_i), \theta_i) + L_{G,x}L_\psi r_\theta + L_{G,\theta}r_\theta \le 0$ for $i = 1, \cdots, N$, then $\forall\theta \in \Theta, \psi(\theta) \in \mathcal{C}_\theta$.*

The complete proof is included in Appendix B.2. Prop. 5.1 establishes sufficient conditions for the trained IPNN to generate IPs for unseen input parameters. These conditions require two key components: (i) the IPNN must achieve feasibility over finite training samples (i.e., $G(\psi(\theta_i), \theta_i) \leq 0$) — a condition readily satisfied in practice, as demonstrated by our empirical results in Sec. 6. The robust penalty loss in (5) effectively minimizes constraint violations for perturbed IP predictions. (ii) to extend feasibility guarantees to the entire input space $\theta \in \Theta$, the IPNN must maintain a *robust margin* for IP prediction that accounts for generalization errors, which can be bounded by Lipschitz conditions (i.e., $L_{G,x} L_\psi r_\theta + L_{G,\theta} r_\theta$).

While direct verification of these Lipschitz constants remains computationally intractable, they provide valuable insights into the relative difficulty of guaranteeing feasibility across different constraint types. Specifically, a smaller covering radius $r_c$ is required for: "thin" constraint sets (i.e., small robust margin) or highly variable constraint geometries (i.e., large $L_{G,\theta}$). For such challenging constraints, a smaller covering radius necessitates a larger number of training samples ($N$), scaling as $\mathcal{O}((\text{diam}(\Theta)/r_c)^d)$ to adequately cover the input space.

We also remark that this condition is based on the Lipschitz-based worst-case analysis, while the empirical experiment shows that IPNN trained over less than 10,000 uniform sample inputs already induces feasible IP prediction under unseen input parameters for high-dimensional problems.

### 5.2. Optimality and Run-time Complexity for Bisection

**Theorem 1.** *Given constraint set $\mathcal{C}_\theta$ under Assumption 1, an infeasible NN prediction $\tilde{x}_\theta$ with bounded error to the optimal solution $x_\theta^*$ as $\|\tilde{x}_\theta - x_\theta^*\| \leq \epsilon_{\text{pre}}$, and valid IP prediction $x_\theta^\circ \in \mathcal{C}_\theta$ produced by IPNN, after executing $K$ steps of bisection shown in Alg. 1. We obtain a solution $\hat{x}_\theta^K$ satisfying the following:*

*(i) it is guaranteed to be feasible, i.e., $\hat{x}_\theta^K \in \mathcal{C}_\theta$;*

*(ii) it has a bounded optimality gap as $\|\hat{x}_\theta^K - x_\theta^*\| \leq 2\epsilon_{\text{pre}} + \mathcal{E}(x_\theta^\circ, \Gamma_\theta) + 2^{-K}(\epsilon_{\text{pre}} + \text{diam}(\mathcal{C}_\theta))$,*

*(iii) the run-time complexity is $\mathcal{O}(KG)$, where $G$ is the complexity of checking the feasibility of a solution.*

The complete proof is included in Appendix B.3. First, given an interior point, the bisection in Alg. 1 consistently returns a feasible solution. The optimality loss of the returned feasible solution is mainly bounded by three factors: the initial NN prediction error for the optimal solution, the eccentricity measure of the predicted IP, and the error due to finite-step bisection. (i) The initial prediction error is typically small, thanks to the NN's universal approximation capabilities. (ii) The eccentricity represents the upper bound of the deviation caused by employing bisection with IPs,

which is mitigated by the Chebyshev center-informed loss function in (5). (iii) The error from finite bisection decreases exponentially with each additional iteration.

The algorithm's run-time complexity, i.e., the number of arithmetic operations, is primarily affected by the number of bisection steps ($K$) and the complexity of checking feasibility at each step ($G$). Such a feasibility checking procedure is also known as *membership oracle*, which has been extensively analyzed in (Mha22; LBGH23). Further, for common convex sets, the bisection projection has a closed-form computation provided in (THH23).

### 5.3. Discussions

**Connection to existing works**: As discussed in Sec. 2, the BP framework is algorithmically related to the homeomorphic projection (LCL23; LCL24) and gauge function-based methods (TZ22b; Mha22; THH23). The following proposition reveals the connection between the BP framework and some existing schemes.

**Proposition 5.2.** *The homeomorphic projection (LCL23) with gauge mapping (TZ22a) is equivalent to bisection projection over a convex set. The Gauge projection (Mha22; THH23) is equivalent to bisection projection with a fixed IP.*

The complete proof is included in Appendix A.2. Thus, the bisection projection framework provides a unified view for some existing projection-analogous approaches over convex sets. Meanwhile, we highlight the theoretical analysis and application scenario for BP works on general compact sets under Assumption 1. It also achieves better performance in feasibility and speedup as shown in Sec. 6.

**Availability and guarantees of IPs.** Several existing NN feasibility approaches rely on IPs, including gauge mapping (TZ22a; TZ22b; LKM23), gauge projections (THH23; KU23; LM23; TVH24), and homeomorphic projections (LCL23; LCL24). While our BP framework similarly utilizes IPs, it advances the state-of-the-art through: general constraints (Sec. 3), eccentricity-based optimality bound (Sec. 4), and IPNN loss design (5). Despite these advances, the guarantees for NN-based IP findings depend on NN training by minimizing penalty-based loss functions. While exact convergence has been established for over-parameterized NNs (SJL18; LZB22), The general convergence analysis remains challenging for practical scenarios involving finite-size networks and non-convex penalties and warrants future exploration.

**Extension to Multiple Interior Points**. The BP framework naturally extends to multiple IPs. Specifically, we can perform bisection from multiple IPs and select the projected point with the minimal projection distance. A detailed discussion of this extension is provided in Appendix A.5.

# 6. Numerical Experiments

We first consider a toy example to visualize the training and testing performance of our framework in Sec. 6.1. We then carry out comprehensive simulations to validate the efficiency of BP against existing methods on various constrained optimization problems in Sec. 6.2. We also demonstrate the efficacy of key design and parameters in the BP framework through sensitivity analysis in Sec. 6.3. The detailed experimental setting and problem formulations are provided in D.

## 6.1. Illustrative Toy Examples

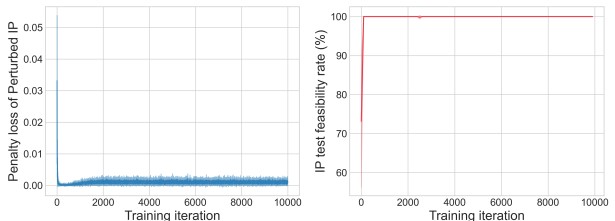

*Figure 3.* Perturbed penalty loss (left) and IPNN prediction feasibility rate (right) during training.

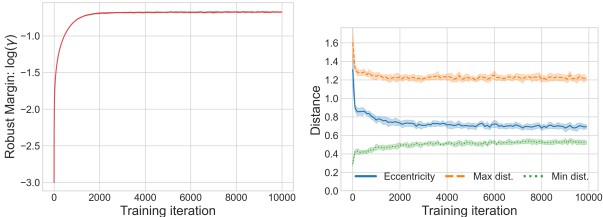

*Figure 4.* Robust margin $\log(\gamma)$ during training (left) and estimated eccentricity (right), calculated by the gap between maximum and minimum IP-to-boundary distances.

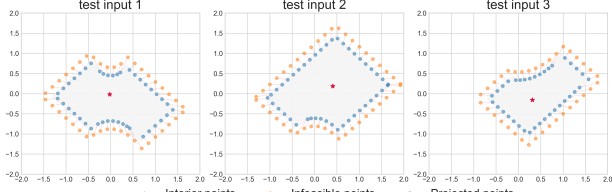

*Figure 5.* BP with IPNN prediction given test input parameters.

To validate that the proposed training algorithm for IPNN can indeed produce low-eccentricity points or approximated Chebyshev centers, we consider a non-convex quadratic constraint set $\mathcal{C}_\theta = \{x \mid x^\top Q_i x + q_i^\top x + b_i \leq 0, i = 1, \cdots, 6\}$, where the input parameter is defined as $\theta = \{Q, q, b\}_{i=1}^6$. We train IPNN over such constraint sets and visualize its training and testing performance. Our experiments yield the following observations: (i) Minimizing the penalty loss successfully improves feasibility over test input parameters, as shown in Fig. 3; (ii) Maximizing the robust margin indeed reduces the eccentricity of IPNN predictions, as shown in Fig. 4; (iii) After training, bisection projection with IPNN-predicted "central" IPs incurs low projection distance, as

shown in Fig. 5. We provide comprehensive visualizations from Fig. 12 to 17 in Appendix E.

## 6.2. NN feasibility for Constrained Optimization

**Dataset**: We apply the BP framework to four benchmark convex problems (QP, convex QCQP, SOCP, and SDP) and two non-convex real-world scenarios, including optimal power flow problems in grid operation (AC-OPF) and joint chance-constrained problems in inventory management (JC-CIM). We follow the established parameter configuration and sampling strategy from available codes in previous works (DRK20; LCL23). We first train an NN predictor to learn the mapping from input parameters to the optimal solutions in existing works (DRK20), where the training and testing data are generated by randomly sampling the input parameter and solve the corresponding optimal solutions through iterative solvers as ground truth (DRK20; LCL23).

**Baselines**: (i) **Optimizer**: for convex optimization, we use MOSEK to solve the optimal solution. For JCC-IM, we adopt its scenario-based approximation and solve it by MOSEK (PAS09); For AC-OPF problems, we adopt PY-POWER as the specialized solver (ZMS11); (ii) **NN**: it directly maps the input parameter to the solution without post-processing; (iii) **WS**: The infeasible prediction of NN is regarded as the warm-start initialization for the iterative solver; (iv) **Proj**: the infeasible predicted solution by NN is processed by orthogonal projection and solved with the iterative solver; (v) **D-Proj**: this is proposed in DC3 (DRK20), which applies gradient descent with equality completion to minimize the constraint violation; (vi) **H-Proj**: the homeomorphic projection are applied to the infeasible predictions (LCL23); (vii) **B-Proj**: we apply bisection in Alg. 1 with predicted IPs to recover the feasibility. To ensure the feasibility of equality constraints, we utilize predict-then-reconstruct techniques (PZCZ20; DRK20), as detailed in Appendix A.4. Note that some baselines shown in Table 1 are not included due to their limited applicability.

Table 2 summarizes our experimental results across six constraint optimization problems, revealing several key insights. The BP framework consistently achieves **100%** feasibility for initially infeasible NN predictions while offering up to four orders of magnitude speedup compared to standard projection approaches, all while maintaining competitive optimality loss. Direct NN outputs cannot guarantee complete feasibility due to prediction errors that may push solutions outside the constraint set. Iterative solver-based methods such as warm-start and orthogonal projection ensure feasibility with minimal optimality loss but incur substantial computational overhead (exceeding 400 seconds for QCQP problems). The gradient-based D-Proj method, though flexible across different constraint sets, fails to guarantee feasibility and exhibits high sensitivity to step size selection. While

*Table 2.* Performance comparison for constrained optimization problems.

| Method | Feasibility rate (%) | Solution opt. (%) | Objective opt. (%) | Pred. cost (s) | Post. cost (s) | Feasibility rate (%) | Solution opt. (%) | Objective opt. (%) | Pred. cost (s) | Post. cost (s) |
|---|---|---|---|---|---|---|---|---|---|---|
| | QP ($n$=400, $d$=100, $n_\text{eq}$=100, $n_\text{ineq}$=100) | | | | | QCQP ($n$=400, $d$=100, $n_\text{eq}$=100, $n_\text{ineq}$=100) | | | | |
| NN | 80.8 | 2.26 | 0.97 | | — | 92.0 | 3.91 | 3.26 | | — |
| NN+WS | 100 | 1.81 | 0.79 | | 5.37 | 100 | 3.58 | 3.01 | | 434 |
| NN+Proj | 100 | 2.26 | 0.97 | | 5.23 | 100 | 3.91 | 3.26 | | 401 |
| NN+D-Proj | 80.9 | 2.26 | 0.97 | 0.0014 | 11.1 | 92.3 | 3.91 | 3.26 | 0.0017 | 7.23 |
| NN+H-Proj | 100 | 16.49 | 15.42 | | 0.763 | 100 | 3.95 | 3.31 | | 0.413 |
| **NN+B-Proj** | **100** | 2.31 | 1.00 | | **0.0160** | **100** | 3.94 | 3.30 | | **0.0162** |
| | SOCP ($n$=400, $d$=100, $n_\text{eq}$=100, $n_\text{ineq}$=100) | | | | | SDP ($n$=40×40, $d$=40, $n_\text{eq}$=40, $n_\text{ineq}$=1) | | | | |
| NN | 89.6 | 1.24 | 0.55 | | — | 58.0 | 2.10 | 2.74 | | — |
| NN+WS | 100 | 1.10 | 0.50 | | 136 | 100 | 1.20 | 1.57 | | 105 |
| NN+Proj | 100 | 1.24 | 0.55 | | 128 | 100 | 2.10 | 2.74 | | 158 |
| NN+D-Proj | 95.5 | 1.24 | 0.55 | 0.0015 | 5.03 | 58.0 | 2.10 | 2.74 | 0.0017 | 236 |
| NN+H-Proj | 100 | 1.25 | 0.56 | | 0.571 | 100 | 31.51 | 33.67 | | 0.958 |
| **NN+B-Proj** | **100** | 1.25 | 0.56 | | **0.0272** | **100** | 2.26 | 2.96 | | **0.0163** |
| | AC-OPF ($n$=476, $d$=400, $n_\text{eq}$=400, $n_\text{ineq}$=1042) | | | | | JCC-IM ($n$=400, $d$=100, $n_\text{eq}$=0, $n_\text{ineq}$=10,100) | | | | |
| NN | 94.0 | 0.15 | 0.001 | | — | 84.5 | 1.94 | 1.40 | | — |
| NN+WS | 100 | 0.14 | 0.001 | | 4.73 | 100 | 1.64 | 1.21 | | 48.3 |
| NN+Proj | 100 | 0.25 | 0.002 | | 14.1 | 100 | 1.94 | 1.40 | | 128 |
| NN+D-Proj | 96.5 | 0.15 | 0.001 | 0.251 | 14.9 | 84.5 | 1.94 | 1.40 | 0.0013 | 81.1 |
| NN+H-Proj | 100 | 0.63 | 0.042 | | 1.32 | 100 | 10.24 | 10.39 | | 0.876 |
| **NN+B-Proj** | **100** | 0.15 | 0.001 | | **1.13** | **100** | 1.98 | 1.45 | | **0.193** |

[1] Evaluation metrics: (i) **Feasibility**: Percentage of feasible solutions among 1,024 test instances, where feasibility requires satisfying equality and inequality constraints within tolerance $\epsilon = 10^{-5}$; (ii) **Optimality**: Mean absolute percentage error (MAPE) between output and optimal solutions for both decision variables and objective values; (iii) **Running-time**: Total inference time comprising NN predictions and post-processing for constraint satisfaction. Iterative solvers are parallelized when computing projections or warm-start solutions.

[2] $d$ and $n$ represent the dimensions for input parameter $\theta$ and output decision $x$, respectively. $n_\text{eq}$ and $n_\text{ineq}$ denote the number of equality and inequality constraints, respectively.

H-Proj also achieves 100% feasibility, it introduces larger optimality gaps and is less efficient than B-Proj, particularly for high-dimensional constraint sets (e.g., SDP), due to complex invertible NN (INN) training and computationally intensive INN calculations during inference.

In summary, across both convex and non-convex constraint sets, BP demonstrates superior performance by achieving perfect feasibility or significantly reduced computational complexity while maintaining comparable optimality loss.

### 6.3. Sensitivity Analysis for BP Framework

We investigate the impact of key components in the BP framework to validate their effectiveness.

**Impacts of $\gamma$ on out-of-sample feasibility (Fig. 6)**:

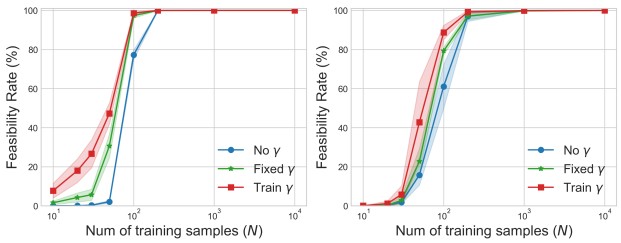

*Figure 6.* Feasibility rates over unseen test instances for QP (left) and JCC-IM (right) under varying training sample sizes $N$.

Prop. 5.1 establishes that increasing both training sample size ($N$) and robust margin ($\gamma$) enhances IPNN's feasibility guarantees on unseen inputs. We empirically validate this relationship by evaluating three IPNN variants: trained with maximizing $\gamma$, trained with fixed $\gamma$, and trained without $\gamma$, under varying training sample sizes. Fig. 6 demonstrates that maximizing the robust margin $\gamma$ improves out-of-sample feasibility (over 1024 test inputs) compared to fixed or zero-margin approaches. These improvements are particularly pronounced under limited training data scenarios, confirming the effectiveness of our proposed Chebyshev center-informed loss function (5).

**Impacts of $\gamma$ on optimality of projected solution (Fig. 7)**:

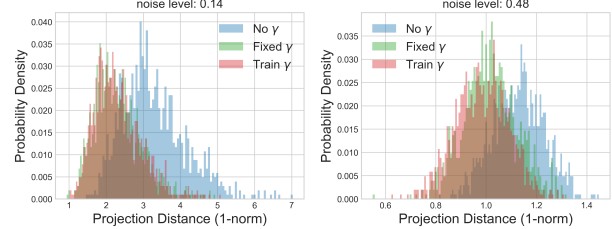

*Figure 7.* Distribution of projection distances for IPNN trained with and without $\gamma$ over QP (left) and JCC-IM (right).

We conducted an ablation study to evaluate the effectiveness of incorporating eccentricity minimization (approximated

by maximizing $\gamma$) on the projection loss. We generated 10,000 infeasible test instances by random Gaussian sampling with varying variance magnitudes (noise levels shown in Fig. 7). For each infeasible point, we applied bisection projection using three IPNN variants: trained with maximizing $\gamma$, trained with fixed $\gamma$, and trained without $\gamma$.

Fig.7 shows the distributions of projection distances (measured between infeasible points and their respective projections). The results demonstrate that our MEIP-informed IPNN (trained with $\gamma$) produces smaller projection distances, confirming the theoretical bounds established in Prop. 4.1. This improved projection quality directly translates to better optimality preservation when correcting infeasible NN predictions.

**Impacts of number of bisection steps (Fig. 8)**:

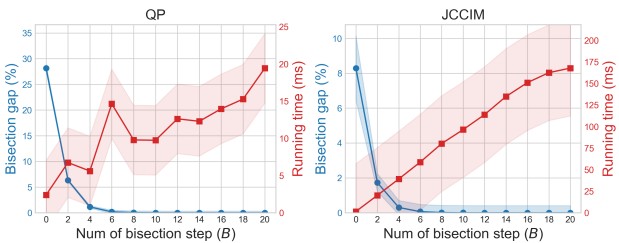

*Figure 8.* Effect of the number of bisection steps ($B$) on optimality gap and running time for QP (left) and JCCIM (right).

We investigated how the number of bisection steps during inference affects both solution quality and computational efficiency. As shown in Fig. 8, and in accordance with Theorem 1, the optimality gap decreases exponentially as the number of bisection steps increases. Concurrently, the running time exhibits linear growth due to the sequential nature of the computation, continuing until reaching the convergence threshold.

Notably, our experiments demonstrate that for real-time applications, only a small number of bisection steps (typically 5-10) are sufficient to achieve well-converged feasible solutions, offering an excellent trade-off between solution quality and computational efficiency. This confirms the practical utility of our approach in time-sensitive decision-making contexts.

**Impacts of bisection stepsize on optimality (Fig. 19)**:

As illustrated in Fig. 2, when multiple intersections exist between the projection ray and the feasible set boundary, the bisection algorithm may converge to any of these intersection points. While this phenomenon did not occur in our benchmark optimization problems, it remains a theoretical concern for constraint sets with complex geometries.

To address potential convergence to boundary points distant from the initial prediction, we may employ a reduced

bisection stepsize (e.g., $\beta = 0.1$ instead of the standard $\beta = 0.5$), which modifies the midpoint calculation to $\alpha_m = \beta \cdot \alpha_l + (1 - \beta) \cdot \alpha_u$. This more conservative approach guides the solution trajectory incrementally from the infeasible prediction toward the feasible boundary, typically resulting in an intersection point closer to the original neural network output.

While this strategy better preserves the quality of initial predictions, it necessitates additional iterations due to the smaller stepsize. This trade-off between solution quality and computational efficiency should be considered based on the specific application requirements and constraint geometry.

**Scalability on constraint dim. and decision dim.**: We further remark on the BP framework's scalability based on large-scale problems in Table 2. For joint chance constraints, where constraint dimension grows linearly with sampled scenarios, iterative solver-based approaches face memory limitations (NS06). For AC-OPF in large-scale power grids, non-linear power balance and branch flow constraints incur high computational complexity for existing solvers (ZMS11). Our bisection methods require only constraint checking per iteration, with GPU-based batch processing further accelerating these calculations.

# 7. Conclusion and Limitation

We introduce **Bisection Projection**, an efficient scheme to project infeasible NN predictions onto general compact constraint sets through bisection. We establish the connection between the eccentricity of interior points (IPs) and projection distance, then employ IPNN to efficiently predict IPs. Our theoretical analysis provides sufficient conditions for IPNN feasibility and proves bounded optimality loss under IP predictions. Extensive simulations demonstrate that bisection projection outperforms existing methods in feasibility and efficiency with comparable optimality.

Our framework has several limitations, suggesting future research directions: (i) extending BP to discrete constraints such as mixed-integer problems for broader applicability, (ii) jointly optimizing interior point selection and bisection trajectory to further reduce optimality gaps, (iii) and exploiting problem-specific structures like symmetry and sparsity to design NN/IPNN to improve training efficiency.

# Acknowledgements

This work is supported in part by a General Research Fund from Research Grants Council, Hong Kong (Project No. 11200223), a Collaborative Research Fund from Research Grants Council, Hong Kong (Project No. C1049-24G), an InnoHK initiative, The Government of the HKSAR, Laboratory for AI-Powered Financial Technologies, and a Shenzhen-Hong Kong-Macau Sci-

ence & Technology Project (Category C, Project No. SGDX20220530111203026). The authors would also like to thank the anonymous reviewers for their helpful comments.

## Code Availability

Code for data generation and model training is available at `Github`.

## Impact Statement

This paper presents work whose goal is to advance the field of Machine Learning. There are many potential societal consequences of our work, none of which we feel must be specifically highlighted here.

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

# Contents

## A. Discussion and Connection of Related Work

### A.1. Availability of Interior Points

We note that there is a line of research on NN feasibility operating under assumptions regarding the availability of IPs.

- **Gauge Mapping** (for Linear Sets): This approach was proposed in (TZ22a; TZ22b), establishing a bijective mapping between a unit cube and a polytope, which can be used as the output layer of neural networks to ensure feasibility over polytopes. The method relies on input-dependent IPs (IPs) and assumes the existence of an affine mapping from input to IP, solving this mapping through semidefinite programming (SDP). However, it does not guarantee the existence of such an affine policy. LOOP-LC (LKM23) directly assumes the existence of an input-invariant IP and solves it by linear constraint residual minimization, but it also does not provide a guarantee or sufficient conditions for such an invariant IP.

- **Gauge Projection** (for Convex Sets): Gauge projection (Mha22) restores solution feasibility by scaling infeasible solutions along rays from a (fixed) offline-computed IP to find feasible solutions at the constraint boundary. This projection technique is differentiable and computationally efficient, which has been successfully integrated into neural network architectures to enforce output feasibility with respect to input-invariant convex sets. Notable implementations include RAYEN (THH23) (supporting linear, quadratic, second-order cone, and linear matrix inequality constraints), ConstraiNet (KU23) (handling linear and quadratic constraints), LOOP-LC 2.0 (LM23) (for linear sets), and the radial projection approach (TVH24) (applicable to several convex cones).

- **Homeomorphic Projection** (for Ball-homeomorphic Sets): It ensures NN solution feasibility over *ball-homeomorphic* constraints by constructing a homeomorphism between the constraint set and a unit ball using invertible NN (INN), allowing efficient projection via bisection (LCL23; LCL24). This approach also relies on a valid INN to map the center of a unit ball to an IP of the constraint set. It provides a sufficient condition for feasibility guarantee (LCL23; LCL24). However, INNs have higher training and inference complexity than regular fully connected neural networks due to their sophisticated architectural design requirements.

- **Bisection Projection** (for General sets): In this work, we consider a more general input-dependent constraint set and propose an efficient bisection projection framework. We first characterize the eccentricity of the IP, which is directly related to the bisection-induced optimality loss. We then relax the minimum-eccentricity IP problem into a tractable Chebyshev center-based formulation and employ another neural network, called IPNN, to efficiently predict IPs under varying inputs for input-dependent constraints. We also provide a sufficient condition for IPNN feasibility similar to (LCL23; LCL24), but applicable to more general sets beyond the ball-homeomorphic ones.

### A.2. Connections to Related Works (Prop. 5.2)

As discussed in Sec. 2, the proposed framework is conceptually related to the homeomorphic projection (LCL23; LCL24) and gauge function based methods (TZ22b; THH23).

**Definition A.1** (Gauge/Minkowski function (BM08)). Let $\mathcal{C} \subset \mathbb{R}^n$ be a compact convex set with a non-empty interior. The Gauge/Minkowski function $\gamma_{\mathcal{C}} : \mathbb{R}^n \times \text{int}(\mathcal{C}) \to \mathbb{R}_+$ is defined as

$$\gamma_{\mathcal{C}}(x, x^\circ) = \inf\{\lambda \geq 0 \mid x \in \lambda(\mathcal{C} - x^\circ)\}, \tag{6}$$

where $x^\circ \in \text{int}(\mathcal{C})$ is an IP of $\mathcal{C}$.

The Gauge function generalizes the concept of a norm. For a set $\mathcal{C}$ that is symmetric about the origin, the gauge function $\gamma_{\mathcal{C}}(x, 0)$ defines a norm. In particular, when $\mathcal{C} = \mathcal{B}_p = \{x \in \mathbb{R}^n \mid |x|p \leq 1\}$ is the unit ball of the $p$-norm, we have $\gamma_{\mathcal{B}_p}(x, 0) = \|x\|_p$.

Based on the gauge function, we can construct the following bijection between two compact convex sets:

**Definition A.2** (Gauge Mapping (TZ22a)). Let $\mathcal{Z}, \mathcal{X} \subset \mathbb{R}^n$ be compact convex sets with IPs $z^\circ \in \text{int}(\mathcal{Z})$ and $x^\circ \in \text{int}(\mathcal{X})$, respectively.

The gauge mapping $\Phi : \mathcal{Z} \to \mathcal{X}$ is defined as:

$$\Phi(z) = \frac{\gamma_{\mathcal{Z}}(z - z^\circ, z^\circ)}{\gamma_{\mathcal{X}}(z - z^\circ, x^\circ)}(z - z^\circ) + x^\circ, \; z \in \mathcal{Z} \tag{7}$$

The inverse mapping $\Phi^{-1} : \mathcal{X} \to \mathcal{Z}$ is given by:

$$\Phi^{-1}(x) = \frac{\gamma_{\mathcal{X}}(x - x^\circ, x^\circ)}{\gamma_{\mathcal{Z}}(x - x^\circ, z^\circ)}(x - x^\circ) + z^\circ, \; x \in \mathcal{X} \tag{8}$$

- In essence, the gauge mapping scales the boundary of a convex set from an IP to another convex set and with translation to its IP.

- When $\mathcal{Z}$ is a unit $p$-norm ball, the gauge mapping is simplified as:

$$\Phi(z) = \frac{\|z\|_p}{\gamma_{\mathcal{C}}(z, x^\circ)} z + x^\circ, \; \forall z \in \mathcal{B}, \qquad \Phi^{-1}(x) = \frac{\gamma_{\mathcal{C}}(x - x^\circ, x^\circ)}{\|x - x^\circ\|_p}(x - x^\circ), \; \forall x \in \mathcal{C}, \tag{9}$$

**Definition A.3** (Gauge Projection (Mha22; THH23)). Let $\mathcal{C} \subset \mathbb{R}^n$ be a compact convex set with $x^\circ \in \text{int}(\mathcal{C})$. For any $x \in \mathbb{R}^n \setminus \mathcal{C}$, the Gauge Projection $\Pi_{\mathcal{C}}^G : \mathbb{R}^n \setminus \mathcal{C} \to \partial\mathcal{C}$ is defined as

$$\hat{x} = \Pi_{\mathcal{C}}^G(x) := x^\circ + \frac{x - x^\circ}{\gamma_{\mathcal{C}}(x - x^\circ, x^\circ)} \in \partial\mathcal{C}, \tag{10}$$

where $\partial\mathcal{C}$ denotes the boundary of $\mathcal{C}$.

**Definition A.4** (Homeomorphic Projection (LCL23)). Let $\mathcal{K} \subset \mathbb{R}^n$ be a compact set that is homeomorphic to the unit ball $\mathcal{B}$. Let $\Phi : \mathcal{B} \to \mathcal{K}$ be a homeomorphism with inverse mapping $\Phi^{-1} : \mathcal{K} \to \mathcal{B}$. For any point $\tilde{x} \in \mathbb{R}^n \setminus \mathcal{K}$, the Homeomorphic Projection $\Pi_{\mathcal{K}}^H : \mathbb{R}^n \setminus \mathcal{K} \to \partial\mathcal{K}$ is defined as:

$$\hat{x} = \Pi_{\mathcal{K}}^H(\tilde{x}) := \Phi(\Pi_{\mathcal{B}}(\Phi^{-1}(\tilde{x}))), \tag{11}$$

where $\Pi_{\mathcal{B}}$ is the Euclidean projection operator.

The following observation reveals the connection between the bisection projection framework and some existing schemes.

**Proposition A.1.** *The homeomorphic projection (LCL23) with gauge mapping (TZ22a) is equivalent to bisection projection over a convex set. The gauge projection (Mha22) is a special case of bisection projection with one fixed IP.*

*Proof.* Let's consider applying the gauge mapping to the homeomorphic projection for a compact convex set. Then we can simplify the homeomorphic projection operator as:

$$\hat{x} = \Phi(\text{Proj}_{\mathcal{B}}(\Phi^{-1}(\tilde{x}))) = \Phi(\text{Proj}_{\mathcal{B}}(\frac{\gamma_{\mathcal{C}}(\tilde{x} - x^\circ, x^\circ)}{\|\tilde{x} - x^\circ\|}(\tilde{x} - x^\circ))) \tag{12}$$

$$= \Phi(\frac{\frac{\gamma_{\mathcal{C}}(\tilde{x} - x^\circ, x^\circ)}{\|\tilde{x} - x^\circ\|}(\tilde{x} - x^\circ)}{\|\frac{\gamma_{\mathcal{C}}(\tilde{x} - x^\circ, x^\circ)}{\|\tilde{x} - x^\circ\|}(\tilde{x} - x^\circ)\|}) = \Phi(\frac{\tilde{x} - x^\circ}{\|\tilde{x} - x^\circ\|}) \tag{13}$$

$$= \frac{\|\frac{\tilde{x} - x^\circ}{\|\tilde{x} - x^\circ\|}\|}{\gamma_{\mathcal{C}}(\frac{\tilde{x} - x^\circ}{\|\tilde{x} - x^\circ\|}, x^\circ)}(\frac{\tilde{x} - x^\circ}{\|\tilde{x} - x^\circ\|}) + x^\circ \tag{14}$$

$$= \frac{\tilde{x} - x^\circ}{\gamma_{\mathcal{C}}(\tilde{x} - x^\circ, x^\circ)} + x^\circ \tag{15}$$

When considering $\tilde{x}$ as an infeasible point, we need scale down $\tilde{x}$ of $\frac{1}{\gamma_{\mathcal{C}}(\tilde{x} - x^\circ, x^\circ)}$ such that the $\frac{1}{\gamma_{\mathcal{C}}(\tilde{x} - x^\circ, x^\circ)}(\tilde{x} - x^\circ) + x^\circ$ will be located in the boundary. Therefore, we take $\alpha^* = \frac{1}{\gamma_{\mathcal{C}}(\tilde{x} - x^\circ, x^\circ)}$, the homeomorphic projection operator is indeed the bisection projection operator in (2). It is also equivalent to the gauge projection or RAYEN methods by its definition in Def. A.3.

$\square$

Thus, the bisection projection framework provides a unified view for some existing projection-analogous approaches over convex sets. Meanwhile, we highlight the theoretical analysis and application scenario for BP works on general compact sets under Assumption 1 beyond those in the existing studies, further exploring the projection-based design and achieving substantially better performance in feasibility, optimality loss, and speedup as shown in Sec. 6.

### A.3. Comparison of Different Centers

Defining the centers of a set is a classic problem in mathematics, which involves various definitions tailored to serve specific purposes. Each definition captures a unique aspect of "centrality" depending on the application or theoretical requirements. Here, as shown in Table 3, we discuss several classic definitions including the proposed minimum-eccentricity interior point (MEIP) in our work.

*Table 3.* Comparison of different definitions of center for a set

| Name | Definition | Description |
|---|---|---|
| MEIP | $x^\circ = \arg\min_{x \in \mathcal{X}} \left( \max_{y \in \partial \mathcal{X}} \|x - y\| - \min_{y \in \partial \mathcal{X}} \|x - y\| \right)$ | Minimizes the discrepancy between the maximum and minimum IP-to-boundary distances. |
| Chebyshev Center | $x^\circ = \arg\max_{x \in \mathcal{X}} \min_{y \in \partial \mathcal{X}} \|x - y\|$ | Maximizes the minimum IP-to-boundary. |
| Circumcircle Center | $x^\circ = \arg\min_{x \in \mathcal{X}} \max_{y \in \partial \mathcal{X}} \|x - y\|$ | Minimizes the maximum IP-to-boundary. |
| Analytical Center | $x^\circ = \arg\max_{x \in \mathcal{X}} \left( \sum_{i=1}^{n_{\text{ineq}}} \log(-g_i(x)) \right)$ | Maximizes the logarithmic barrier of the inequality residuals ($g_i(x) \le 0$). |
| Max-residual Center | $\max_{x^\circ, t \ge 0} t \quad \text{s.t. } g_i(x^\circ) + t \le 0, \ i = 1, \ldots, n_{\text{ineq}}$ | Maximize the constraint residual to find a "central" IP. |
| Centroid | $x^\circ = \frac{1}{n} \sum_{i=1}^{n} x_i$ | Calculates the average position of all points in the set. |
| Barycenter | $x^\circ = \sum_{i=1}^{n} w_i x_i / \sum_{i=1}^{n} w_i$ | Calculates the weighted average position of all points in the set, where each point has an associated weight $w_i$. |

- **Geometric**: The MEIP, Chebyshev Center, and Circumcircle Center focus on geometric properties of sets, specifically distances to the boundary. We propose the MEIP for bisection operation, justified by the performance guarantee in Prop. 4.1.

- **Barrier**: The Analytical Center and Max-residual Center use optimization techniques (BBV04; THH23) to find a "central" IP within a feasible region. This approach is crucial for solving linear and nonlinear programming problems. However, maximizing the log-residual or residual directly of the inequality function does not directly reflect the point-to-boundary distance for general constraint sets, which may result in a large deviation for the bisection operation.

- **Statistical**: The Centroid and Barycenter represent statistical approaches to defining centrality by calculating averages of point sets. The Centroid computes a simple arithmetic mean, suitable for applications in statistics and machine learning. The Barycenter incorporates weights, allowing for differentiated influence among points. Despite their simplicity, these centers are not necessarily interior points for general non-convex sets $\mathcal{X}$, making them unsuitable for our bisection operation.

### A.4. Tackling Equality Constraint

Consider the following constraint set $\mathcal{C}_\theta$ defined by both inequality and equality constraints:

$$\mathcal{C}_\theta = \{x \in \mathbb{R}^n \mid h(x, \theta) = 0, \ g(x, \theta) \leq 0\}, \tag{16}$$

where the functions $h(\cdot, \cdot) : \mathbb{R}^{n+d} \to \mathbb{R}^{n_{\text{eq}}}$ and $g(\cdot, \cdot) : \mathbb{R}^{n+d} \to \mathbb{R}^{n_{\text{ineq}}}$ are continuous with respect to $x$ and $\theta$. For simplicity, we use $h_\theta(\cdot) = h(\cdot, \theta)$.

Assuming the equality constraint maintains a constant rank:

$$\text{rank}(\text{J}_{h_\theta}(x)) = r, \quad \forall \theta \in \Theta \text{ and } \forall x \in \mathcal{C}_\theta, \tag{17}$$

This condition implies that $\mathcal{C}_\theta$ has a Euclidean dimension[5] of $n - r$, as per the *Constant-Rank Level Set Theorem* (Lee13).

In simpler terms, we can utilize a subset of decision variables $x_1 \in \mathbb{R}^{n-r}$ and reconstruct the complete set of decision variables $[x_1, x_2] \in \mathbb{R}^n$ by solving $x_2 = \phi_\theta(x_1)$, such that $h_\theta([x_1, \phi_\theta(x_1)]) = 0$. Note that such a parametrization are not necessarily held globally for non-linear equality constraints. This method of reconstruction, which ensures the feasibility of the equality constraint, is extensively used in optimization literature (Aba69; PZC19; ZB20; DRK20; LCL23; THH23; LM23; DWDS23).

We then denote the reduced constraint set as

$$\mathcal{C}_\theta^s = \{x \in \mathbb{R}^{n-r} \mid g([x_1, \phi_\theta(x_1)], \theta) \leq 0\} \tag{18}$$

This set $\mathcal{C}_\theta^s$ is not only equivalent to the original constraint set $\mathcal{C}_\theta$ but also homeomorphic to it, implying a one-to-one, continuous, and bicontinuous correspondence between the two sets. The forward and inverse mappings of this homeomorphism are described by the following transformations:

$$[x_1, x_2] \in \mathcal{C}_\theta \to x_1 \in \mathcal{C}_\theta^s, \tag{19}$$

$$x_1 \in \mathcal{C}_\theta^s \to [x_1, \phi_\theta(x_1)] \in \mathcal{C}_\theta. \tag{20}$$

Let's consider two examples to illustrate this equality completion/reconstruction process:

**Linear equality constraint**

Let's consider an equality constraint defined as $\{x \in \mathbb{R}^n \mid Ax = \theta, A \in \mathbb{R}^{r \times n}, \theta \in \mathbb{R}^r\}$, where $x$ is the decision variable and $\theta$ is the input parameter. We can assume, without loss of generality, that the rank of matrix $A$ is $\text{rank}(A) = r$.

To facilitate the reconstruction process, we partition the decision variable $x$ into two groups: $x_1 \in \mathbb{R}^{n-r}$ and $x_2 \in \mathbb{R}^r$. Accordingly, we also partition matrix $A$ into $A = [A_1, A_2]$, where $A_1 \in \mathbb{R}^{r \times (n-r)}$ and $A_2 \in \mathbb{R}^{r \times r}$. Hence, the equality constraint can be represented as $A_1 x_1 + A_2 x_2 = \theta$. The reconstruction process indicates that we can determine $x_2$ using only the subset of variables $x_1$, with the explicit relationship given by:

$$x_2 = \phi_\theta(x_1) = A_2^{-1}(\theta - A_1 x_1). \tag{21}$$

---

[5]If an open set $\mathcal{X}$ is Euclidean of dimension, then every point $x \in \mathcal{X}$ has a neighborhood that is homeomorphic to an open subset of $\mathbb{R}^n$ (Lee13).

Here, we choose the partition of $x_1$ and $x_2$ such that $A_2$ has the full rank of $r$.

The relevant Jacobian matrix for back-propagation in this context is:

$$\mathrm{J}_{\phi_\theta}(x_1) = -A_2^{-1} A_1. \tag{22}$$

**Non-linear equality constraint**

For a non-linear equality constraint defined as $\{x \in \mathbb{R}^n \mid h(x, \theta) = 0, \theta \in \mathbb{R}^d, h : \mathbb{R}^{n+d} \to \mathbb{R}^r\}$, we partition the decision variable into $x_1 \in \mathbb{R}^{n-r}$ and $x_2 \in \mathbb{R}^r$ in a similar fashion to the linear case. Under the assumption that the Jacobian matrix of $h$ with respect to $x_2$ has a constant rank, the completion function $\phi_\theta$ is well-defined and satisfies:

$$h([x_1, \phi_\theta(x_1)], \theta) = 0. \tag{23}$$

To solve for $\phi_\theta(x_1)$ when $h$ is non-linear, we can employ an iterative technique such as Newton's method. The necessary Jacobian matrix for back-propagation can be computed using the *Implicit Function Theorem*, which provides the derivative of the implicitly defined function $\phi_\theta$. The Jacobian matrix is given by:

$$\mathrm{J}_{\phi_\theta}(x_1) = - \mathrm{J}_{h_\theta}^{-1}(x_2) \, \mathrm{J}_{h_\theta}(x_1). \tag{24}$$

Note that $\phi_\theta$ for such a non-linear constraint may not be single-valued globally and depends on the initial value for the iterative algorithm, which may bring potential convergence issues.

In conclusion, reconstruction techniques utilizing equality constraints allow for a reduction in the dimensionality of the decision variable space. By modeling only a subset of the decision variables, we can focus on the inequality constraints and use the equality constraints to define the remaining variables implicitly. This process is differentiable, making it suitable for integration into the training of machine learning models, hence providing a powerful tool for incorporating equality constraints into such models (Aba69; PZC19; PCZL22; DRK20; DWDS23). For the implementation issues, we follow the established procedures in previous works (DRK20; LCL23), where the partial variables for convex problems are randomly sampled, and the ones for AC-OPF problems are strategically selected.

## A.5. Extension BP to Multiple Interior Points

The bisection method can be executed in batch for multiple interior points $X_{\theta,m}^\circ := \{x_{\theta,k}\}_{k=1}^m \subset \mathcal{C}_\theta$, and we select the projected point as the one with minimum deviation, defined as:

$$\hat{x}_\theta = \mathrm{BP}(\tilde{x}_\theta, X_{\theta,m}^\circ) \triangleq \arg\min_{\hat{x}_{\theta,k}} \{\|\hat{x}_{\theta,k} - \tilde{x}_\theta\|\}, \tag{25}$$

where $\hat{x}_{\theta,k} = \mathrm{BP}(\tilde{x}_\theta, x_{\theta,k}^\circ)$ is the returned feasible point by bisection w.r.t. the $k$-th IP $x_{\theta,k}^\circ \in X_{\theta,m}^\circ$.

Similarly, we can define the eccentricity of a set of IPs, crucial for bounding the bisection-induced projection distance.

**Definition A.5** (Eccentricity of IPs). For a compact set $\mathcal{X}$ satisfying Assumption 1 with non-empty interior, the eccentricity of a set of IPs $X_m^\circ := \{x_k^\circ\}_{k=1}^m \subset \mathcal{X}$ with respect to a compact subset of boundary $\Gamma \subseteq \partial \mathcal{X}$ is defined as:

$$\mathcal{E}(X_m^\circ, \Gamma) \triangleq \max_{y \in \Gamma} \|\mathrm{d}(y, X_m^\circ)\| - \min_{y \in \Gamma} \|\mathrm{d}(y, X_m^\circ)\|, \tag{26}$$

where $\mathrm{d}(y, X_m^\circ) = \min_{1 \le k \le m} \{\|y - x_k^\circ\|\}$ is the point-to-set distance.

Next, we establish the connection between eccentricity and the bisection-induced projection distance.

**Proposition A.2.** *Let* $\tilde{x}_\theta = F(\theta)$ *be an infeasible NN prediction with bounded prediction error as* $\|F(\theta) - x_\theta^*\| \le \epsilon_{\mathrm{pre}}$*;* $\hat{x}_\theta = \mathrm{BP}(\tilde{x}_\theta, X_{\theta,m}^\circ)$ *be the projected solution with $m$ interior points $X_{\theta,m}^\circ \subset \mathcal{C}_\theta$; Then, the worst-case projection distance is upper bounded as:*

$$\max_{\tilde{x}_\theta \in \mathcal{B}(x_\theta^*, \epsilon_{\mathrm{pre}})} \|\tilde{x}_\theta - \mathrm{BP}(\tilde{x}_\theta, X_{\theta,m}^\circ)\| \le \epsilon_{\mathrm{pre}} + \mathcal{E}(X_{\theta,m}^\circ, \Gamma_\theta), \tag{27}$$

*where $\mathcal{B}(x_\theta^*, \epsilon_{\mathrm{pre}})$ represents the NN prediction region, enclosing all infeasible NN predictions with prediction error $\epsilon_{\mathrm{pre}}$, and $\Gamma_\theta = \{\mathrm{BP}(\tilde{x}_\theta, X_{\theta,m}^\circ), \forall \tilde{x}_\theta \in \mathcal{B}(x_\theta^*, \epsilon_{\mathrm{pre}}) \setminus \mathcal{C}_\theta\}$ defines a subset of the constraint boundary containing all projected NN solutions from the NN infeasibility region.*

The proof is similar to the single IP case and presented in the next section.

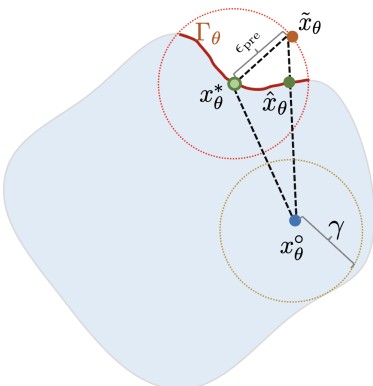

*Figure 9.* A geometric illustration of eccentricity and the proof.

# B. Proof for Main Results

### B.1. Proof for Bisection-induced Projection Distance (Prop. 4.1)

*Proof.* Without loss of generality, we assume that the optimal solution $x_\theta^* \in \partial\mathcal{C}_\theta$ lies on the boundary of the constraint set, implying the existence of active constraints at optimality. This is a standard assumption in constrained optimization theory (NW99).

For an infeasible solution $\tilde{x}_\theta \in \mathcal{B}(x_\theta^*, \epsilon_{\text{pre}})$, recall the definition of the projected solution $\hat{x}_\theta$ as:

$$\hat{x}_\theta \triangleq \mathrm{BP}(\tilde{x}_\theta, x_\theta^\circ) = \alpha^* \cdot (\tilde{x}_\theta - x_\theta^\circ) + x_\theta^\circ, \tag{28}$$

where $\alpha^* \in [0,1]$ leads to $\hat{x}_\theta \in \partial\mathcal{C}_\theta$.

**Prediction-agnostic bound**:

Then, the projection distance for an infeasible prediction $\tilde{x}_\theta$ is bounded as:

$$\|\tilde{x}_\theta - \hat{x}_\theta\| \overset{(a)}{=} \|\tilde{x}_\theta - x_\theta^\circ\| - \|\hat{x}_\theta - x_\theta^\circ\| \tag{29}$$

$$\overset{(b)}{\leq} \|\tilde{x}_\theta - x_\theta^*\| + \|x_\theta^* - x_\theta^\circ\| - \|\hat{x}_\theta - x_\theta^\circ\| \tag{30}$$

$$\overset{(c)}{\leq} \epsilon_{\text{pre}} + \|x_\theta^* - x_\theta^\circ\| - \|\hat{x}_\theta - x_\theta^\circ\| \tag{31}$$

$$\overset{(d)}{\leq} \epsilon_{\text{pre}} + \max_{x \in \Gamma_\theta} \|x - x_\theta^\circ\| - \min_{x \in \Gamma_\theta} \|x - x_\theta^\circ\| \tag{32}$$

$$\overset{(e)}{=} \epsilon_{\text{pre}} + \mathcal{E}(x_\theta^\circ, \Gamma_\theta) \tag{33}$$

$$\leq \epsilon_{\text{pre}} + \mathcal{E}(x_\theta^\circ, \partial\mathcal{C}_\theta) \tag{34}$$

Equality $(a)$ is by three points, $\tilde{x}_\theta$, $x_\theta^\circ$, and $\hat{x}_\theta$, exist in the same straight line. Inequality $(b)$ is by the triangle inequality with auxiliary point $x_\theta^*$. Inequality $(c)$ is by $\tilde{x}_\theta \in \mathcal{B}(x_\theta^*, \epsilon_{\text{pre}})$. Inequality $(d)$ is by taking the maximum and minimum point over local boundary $\Gamma_\theta = \{\mathrm{BP}(\tilde{x}_\theta, x_\theta^\circ), \forall \tilde{x}_\theta \in \mathcal{B}(x_\theta^*, \epsilon_{\text{pre}}) \setminus \mathcal{C}_\theta\}$. Equality $(e)$ is by the definition of eccentricity in Def. 4.1.

**Prediction-aware bound**: Based on (31), the projection distance for an infeasible prediction $\tilde{x}_\theta$ is bounded as:

$$\|\tilde{x}_\theta - \hat{x}_\theta\| \leq \epsilon_{\text{pre}} + \|x_\theta^* - x_\theta^\circ\| - \|\hat{x}_\theta - x_\theta^\circ\| \tag{35}$$

$$\leq \epsilon_{\text{pre}} + \|x_\theta^* - x_\theta^\circ\| - \min_{x \in \partial\mathcal{C}_\theta} \|x - x_\theta^\circ\| \tag{36}$$

where $\epsilon_{\text{int}} = \|x_\theta^* - x_\theta^\circ\|$ denotes the distance between the IP prediction and the optimal solution, and $\gamma = \min_{x \in \partial\mathcal{C}_\theta} \|x - x_\theta^\circ\|$ denotes minimum point-to-boundary distance.

**Proof for Proposition A.2:**

For the multiple IPs setting $X_{\theta,m}^{\circ} = \{x_{\theta,k}^{\circ}\}_{k=1}^{m} \subset \mathcal{C}_{\theta}$, we derive the upper bound as follows:

$$\min_{1 \leq k \leq m} \|\tilde{x}_{\theta} - \hat{x}_{\theta,k}\| \overset{(a)}{\leq} \min_{1 \leq k \leq m} \{\epsilon_{\text{pre}} + \|x_{\theta}^{*} - x_{\theta,k}^{\circ}\| - \|\hat{x}_{\theta,k} - x_{\theta,k}^{\circ}\|\} \tag{37}$$

$$\overset{(b)}{\leq} \epsilon_{\text{pre}} + \min_{1 \leq k \leq m} \|x_{\theta}^{*} - x_{\theta,k}^{\circ}\| - \min_{1 \leq k \leq m} \|\hat{x}_{\theta,k} - x_{\theta,k}^{\circ}\| \tag{38}$$

$$\overset{(c)}{\leq} \epsilon_{\text{pre}} + \max_{x \in \Gamma_{\theta}} \min_{1 \leq k \leq m} \|x - x_{\theta,k}^{\circ}\| - \min_{x \in \Gamma_{\theta}} \min_{1 \leq k \leq m} \|x - x_{\theta,k}^{\circ}\| \tag{39}$$

$$\overset{(d)}{=} \epsilon_{\text{pre}} + \mathcal{E}(X_{\theta,m}^{\circ}, \Gamma_{\theta}) \tag{40}$$

Inequality $(a)$ is by the bound for the single-IP setting above. Inequality $(b)$ is by the minimization of the joint term, which is smaller than the minimization separately. Inequality $(c)$ is by taking the maximum and minimum point over the local boundary $\Gamma_{\theta} = \partial \mathcal{C}_{\theta} \cap \{\text{BP}(\tilde{x}_{\theta}, X_{\theta,m}^{\circ}), \forall \tilde{x}_{\theta} \in \mathcal{B}(x_{\theta}^{*}, \epsilon_{\text{pre}}) \setminus \tilde{x}_{\theta} \notin \mathcal{C}_{\theta}\}$

Thus, we complete the proof as follows:

$$\max_{\tilde{x}_{\theta} \in \mathcal{B}(x_{\theta}^{*}, \epsilon_{\text{pre}})} \min_{1 \leq k \leq m} \|\tilde{x}_{\theta} - \hat{x}_{\theta,k}\| \leq \epsilon_{\text{pre}} + \mathcal{E}(X_{\theta,m}^{\circ}, \Gamma_{\theta}) \tag{41}$$

$\square$

## B.2. Proof for Feasibility Guarantee in (Prop. 5.1)

*Proof.* Since $\mathcal{D}$ is an $r_{\theta}$-covering dataset for $\Theta$ as $\Theta \subseteq \bigcup_{i=1}^{N} \mathcal{B}(\theta_{i}, r_{\theta})$, for any $\theta \in \Theta$, there exists at least one $\theta_{i} \in \mathcal{D}$ such that:

$$\|\theta - \theta_{i}\| \leq r_{\theta} \tag{42}$$

Given that $\psi$ is $L_{\psi}$-Lipschitz continuous over $\Theta$, we have:

$$\|\psi(\theta) - \psi(\theta_{i})\| \leq L_{\psi}\|\theta - \theta_{i}\| \leq L_{\psi}r_{\theta}, \tag{43}$$

Since constraint violation function $G$ is $L_{G,x}$-Lipschitz in $x$ and $L_{G,\theta}$-Lipschitz in $\theta$, we can bound the change in $G$ due to perturbations in both arguments:

$$G(\psi(\theta), \theta) \leq G(\psi(\theta_{i}), \theta_{i}) + L_{G,x}\|\psi(\theta) - \psi(\theta_{i})\| + L_{G,\theta}\|\theta - \theta_{i}\|$$
$$\leq G(\psi(\theta_{i}), \theta_{i}) + L_{G,x}L_{\psi}r_{\theta} + L_{G,\theta}r_{\theta}.$$

Suppose for each $\theta_{i} \in \mathcal{D}$, the training requirement is satisfied as: $G(\psi(\theta_{i}), \theta_{i}) + L_{G,x}L_{\psi}r_{\theta} + L_{G,\theta}r_{\theta} \leq 0$, then we have for any $\theta \in \Theta$, the constraint violation can be bounded as $G(\psi(\theta), \theta) \leq 0$.

$\square$

## B.3. Proof for Optimality Gap and Run-time Complexity (Theorem 1)

*Proof.* First, the feasibility of the solution returned through bisection is guaranteed due to the bisection trajectory connecting an infeasible point and an interior point, which must intersect the constraint boundary. Thus, the bisection algorithm can always find a feasible solution by scaling down the infeasible solution along the line segment. We remark that for general non-convex sets, the line segment between an infeasible point and an interior point may intersect the constraint boundary multiple times, causing our bisection algorithm to converge to one of the multiple feasible solutions.

Let $\hat{x}_{\theta} \in \partial \mathcal{C}_{\theta}$ be the converged boundary feasible solution given infinite bisection with an interior point $x_{\theta}^{\circ}$.

We divide the optimality gap by the following three terms:

$$\|\hat{x}_{\theta}^{K} - x_{\theta}^{*}\| \leq \underbrace{\|x_{\theta}^{*} - \tilde{x}_{\theta}\|}_{\text{prediction error}} + \underbrace{\|\tilde{x}_{\theta} - \hat{x}_{\theta}\|}_{\text{projection error}} + \underbrace{\|\hat{x}_{\theta} - \hat{x}_{\theta}^{K}\|}_{\text{bisection error}} \tag{44}$$

The prediction error is determined by the provided NN predictor, and we denoted it as $\epsilon_{\text{pre}} = \sup_{\theta \in \Theta}\{\|F(\theta) - x_\theta^*\|\}$, where $F(\cdot)$ is the NN predictor to predict the optimal solution.

The projection error from bisection-projection, as proved in Proposition 4.1, can be bounded by the eccentricity-related term as:

$$\|\tilde{x}_\theta - \hat{x}_\theta\| \leq \max_{y \in \mathcal{B}(x_\theta^*, \epsilon_{\text{pre}})} \|\tilde{x}_\theta - \hat{x}_\theta\| \tag{45}$$

$$\leq \epsilon_{\text{pre}} + \mathcal{E}(x_\theta^\circ, \Gamma_\theta) \tag{46}$$

Since $\hat{x}_\theta$ is the converged boundary feasible solution under the bisection algorithm, the bisection error induced by finite step iteration can be derived as:

$$\|\hat{x}_\theta - \hat{x}_\theta^K\| \leq \|\alpha^* \cdot (\tilde{x}_\theta - x_\theta^\circ) + x_\theta^\circ - (\alpha^K \cdot (\tilde{x}_\theta - x_\theta^\circ) + x_\theta^\circ)\| \tag{47}$$

$$= (\alpha^* - \alpha^K)\|\tilde{x}_\theta - x_\theta^\circ\| \tag{48}$$

$$\leq 2^{-K}\|\tilde{x}_\theta - x_\theta^* + x_\theta^* - x_\theta^\circ\| \tag{49}$$

$$\leq 2^{-K}(\|\tilde{x}_\theta - x_\theta^*\| + \|x_\theta^* - x_\theta^\circ\|) \tag{50}$$

$$\leq 2^{-K}(\epsilon_{\text{pre}} + D) \tag{51}$$

where $D = \text{diam}(\mathcal{C}_\theta)$ denote the diameter of a compact set.

Combining the three terms together, we have:

$$\|\hat{x}_\theta^K - x_\theta^*\| \leq 2\epsilon_{\text{pre}} + \mathcal{E}(x_\theta^\circ, \Gamma_\theta) + 2^{-K}(\epsilon_{\text{pre}} + D) \tag{52}$$

The complexity of executing the bisection algorithm involves the iteration steps, the number of IPs, and the complexity of verifying the inequality constraints at each iteration as $G$. For example, if the inequality constraint $g_i(x, \theta)$ is a linear function for all $i = 1, \cdots, n_{\text{ineq}}$, then $G = n \cdot n_{\text{ineq}}$. In contrast, iterative algorithms such as interior point methods have a complexity of $\mathcal{O}((n + n_{\text{ineq}})^3)$ at each iteration due to the matrix inversion operation.

$\square$

# C. Training for IPNN

## C.1. Prediction-agnostic training

In scenarios where prior information about the optimal solution or a trained neural network (NN) predictor is unavailable, our objective is to minimize the worst-case projection distance induced by bisection under any NN predictor. Guided by Proposition 4.1, we aim to minimize the eccentricity relative to the constraint boundary, denoted as $\mathcal{E}(\psi(\theta), \partial\mathcal{C}_\theta)$.

**Definition C.1** (Prediction-agnostic MEIP). For a compact set $\mathcal{C}_\theta$ with a non-empty interior, the minimum eccentricity IP is defined as the solution of the following problem:

$$\min_{x_\theta^\circ} \mathcal{E}(x_\theta^\circ, \partial\mathcal{C}_\theta) \triangleq \max_{y \in \partial\mathcal{C}_\theta} \|y - x_\theta^\circ\| - \min_{y \in \partial\mathcal{C}_\theta} \|y - x_\theta^\circ\|. \tag{53}$$

To facilitate the minimization of the MEIP, we adopt the *Chebyshev Center* as a relaxation. The Chebyshev Center provides a robust central point within the constraint set by maximizing the minimal distance from the center to the boundary.

**Definition C.2** (Chebyshev center). For a compact set $\mathcal{C}_\theta$ with a non-empty interior, the Chebyshev Center is defined as:

$$\max_{x_\theta^\circ} \gamma, \quad \text{s.t.} \ \mathcal{B}(x_\theta^\circ, \gamma) \subseteq \mathcal{C}_\theta \tag{54}$$

Building upon this formulation, we derive the Prediction-Agnostic loss function, which aims to encourage the NN predictor $\psi(\theta)$ to lie within the feasible set while maximizing the robust margin $\gamma$.

$$\mathcal{L}(\psi(\theta), \gamma) = \mathbb{E}_u\left[\text{P}(x_\theta^\circ + \gamma u, \theta)\right] - \lambda \cdot \log(\gamma) \tag{55}$$

where $P(x, \theta)$ represents the penalty function for constraint violations, $u$ is sampled from unit ball to perturb the center $x_\theta^\circ$, $\gamma$ is a variable representing the robust margin to be maximized, and $\lambda$ is a positive coefficient to balance different loss terms. The first term ensures that perturbed points within the ball $\mathcal{B}(x_\theta^\circ, \gamma)$ satisfy the constraints, while the second term encourages maximizing the margin $\gamma$.

### C.2. Prediction-aware training

In contrast, when a trained NN predictor $F(\cdot)$ or a dataset of optimal solutions $(\theta, x_\theta^*)$ is available, the objective shifts to identifying interior points with minimized eccentricity within a local region of the constraint boundary. Given the bound on the bisection-induced projection distance from Equation (36):

$$\|\tilde{x}_\theta - \hat{x}_\theta\| \leq \epsilon_{\text{pre}} + \|x_\theta^* - x_\theta^\circ\| - \min_{x \in \partial \mathcal{C}_\theta} \|x - x_\theta^\circ\| \tag{56}$$

We define the following Prediction-Aware MEIP:

**Definition C.3** (Prediction-aware MEIP)**.**

$$\min \|x_\theta^* - x_\theta^\circ\| \tag{57}$$

$$\text{s.t.} \left\{ \begin{array}{ll} \max_{x_\theta^\circ} & \gamma \\ \text{s.t.} & \mathcal{B}(x_\theta^\circ, \gamma) \subseteq \mathcal{C}_\theta \end{array} \right\} \tag{58}$$

To effectively minimize the bisection-induced projection distance when the optimal solution dataset $x^*$ or NN predictor $F(\cdot)$ is available, we design the following Prediction-Aware loss function:

$$\mathcal{L}(\psi(\theta), \gamma) = \mathbb{E}_u \left[ P(x_\theta^\circ + \gamma u, \theta) \right] - \lambda \cdot \log(\gamma) + \beta \|x_\theta^\circ - x_\theta^*\| \tag{59}$$

- The first two terms are analogous to the Prediction-Agnostic loss, promoting feasibility and robustness.

- The third term penalizes the deviation of the IP from the known optimal solution or initial NN prediction $x_\theta^* = F(\theta)$, enhancing optimality.

- $\beta$ is a positive coefficient that balances the trade-off between maximizing the robust margin $\gamma$ and minimizing the distance to the optimal solution $x_\theta^*$.

- Further, to reduce training complexity, we can initialize the IPNN with the one trained NN solution predictor, then follow the loss to fine-tune the IPNN to find interior points.

Notably, when $\gamma = 0$, the loss function reduces to regular supervised training with a constraint violation penalty, as commonly used in NN-based constrained optimization solvers (PZC19; DRK20). By maximizing the robust margin $\gamma$, the training process follows the preventive/adversarial approach (ZPC+20), seeking an NN predictor that maintains feasibility by keeping distance from constraint boundaries. Rather than directly using such a feasible IP with its larger optimality gap, we apply bisection between the IP and an infeasible but near-optimal NN solution, effectively balancing the feasibility-optimality trade-off.

## D. Data and Experiment Setting

### D.1. Formulation for Optimization Problems

We test the Bisection Projection framework for four constrained optimization problems, including two convex optimization problems and two real-world non-convex problems. we follow the established procedures from available codes in previous works (DRK20; LCL23), where parameter configuration and sampling strategy are publicly available.

#### D.1.1. CONVEX PROBLEM FORMULATION

The Quadratic Program (QP) is a fundamental optimization problem where the objective function is quadratic and the constraints are linear. The QP problem can be formulated as:

$$\mathbf{QP}: \quad \underset{\mathbf{x} \in \mathbb{R}^n}{\text{minimize}} \quad \frac{1}{2}\mathbf{x}^\top Q\mathbf{x} + \mathbf{p}^\top \mathbf{x} \tag{60}$$

$$\text{subject to} \quad G\mathbf{x} \le \mathbf{h}, \tag{61}$$

$$A\mathbf{x} = \boldsymbol{\theta}, \tag{62}$$

where: $Q \in \mathbb{S}_{++}^n$ is a positive definite matrix, ensuring the convexity of the objective function, $\mathbf{p} \in \mathbb{R}^n$ is a vector of linear coefficients, $A \in \mathbb{R}^{n_{eq} \times n}$ is a matrix defining equality constraints, $G \in \mathbb{R}^{n_{ineq} \times n}$ is a matrix defining inequality constraints, $\mathbf{h} \in \mathbb{R}^{n_{ineq}}$ is a vector specifying the upper bounds for the inequality constraints, and $\theta \in \mathbb{R}^{n_{eq}}$ is a vector specifying the right-hand side of the equality constraints.

The Convex QCQP extends the QP by including quadratic constraints. The Convex QCQP problem is given by:

$$\textbf{Convex QCQP}: \quad \underset{\mathbf{x} \in \mathbb{R}^n}{\text{minimize}} \quad \frac{1}{2}\mathbf{x}^\mathsf{T} Q\mathbf{x} + \mathbf{p}^\mathsf{T}\mathbf{x} \tag{63}$$

$$\text{subject to} \quad \mathbf{x}^\mathsf{T} H_i \mathbf{x} + \mathbf{g}_i^\mathsf{T}\mathbf{x} \le h_i, \ i = 1, \ldots, n_{ineq}, \tag{64}$$

$$A\mathbf{x} = \boldsymbol{\theta}, \tag{65}$$

where each $H_i \in \mathbb{S}_{++}^n$ is a positive definite matrix corresponding to the $i$-th quadratic constraint, $\mathbf{g}_i \in \mathbb{R}^n$ is a vector of linear coefficients for the quadratic constraints, and $h_i \in \mathbb{R}$ represents the upper bound for the $i$-th quadratic constraint.

The SOCP is a convex optimization problem that generalizes linear and quadratic programs by allowing conic constraints. A SOCP problem is formulated as follows:

$$\textbf{SOCP}: \quad \underset{\mathbf{x} \in \mathbb{R}^n}{\text{minimize}} \quad \frac{1}{2}\mathbf{x}^\mathsf{T} Q\mathbf{x} + \mathbf{p}^\mathsf{T}\mathbf{x} \tag{66}$$

$$\text{subject to} \quad \|G_i\mathbf{x} + \mathbf{h}_i\|_2 \le \mathbf{c}_i^\mathsf{T}\mathbf{x} + d_i, \ i = 1, \ldots, n_{ineq}, \tag{67}$$

$$A\mathbf{x} = \boldsymbol{\theta}, \tag{68}$$

where $G_i \in \mathbb{R}^{m \times n}$ and $\mathbf{h}_i \in \mathbb{R}^m$ define the second-order cone, $\mathbf{c}_i \in \mathbb{R}^n$ and $d_i \in \mathbb{R}$ are the coefficients and scalar terms of the conic constraints, respectively.

The Semidefinite Program (SDP) is an optimization problem where the goal is to minimize a linear objective function subject to semidefinite constraints. The standard formulation of an SDP is given by:

$$\textbf{SDP}: \quad \underset{X \in \mathbb{S}^n}{\text{minimize}} \quad \text{tr}(CX) \tag{69}$$

$$\text{subject to} \quad X \succeq 0, \tag{70}$$

$$\text{tr}(A_i X) = b_i, \quad i = 1, \ldots, n_{eq}, \tag{71}$$

where $X \in \mathbb{S}^n$ is the symmetric matrix variable, $C \in \mathbb{R}^{n \times n}$ is a given symmetric matrix of coefficients for the objective function, $\textbf{tr}(\cdot)$ is the trace of a matrix, $A_i \in \mathbb{R}^{n \times n}$ are given symmetric matrices that define the equality constraints, $b_i \in \mathbb{R}$ are the given scalars that specify the right-hand side of the equality constraints, and $n_{eq}$ is the number of equality constraints. We discuss the penalty design for PSD constraint in Appendix D.3.

### D.1.2. JOINT CHANCE CONSTRAINED INVENTORY MANAGEMENT (JCC-IM)

We consider the Joint Chance-Constrained Inventory Management (JCC-IM) problem, which seeks to optimize inventory levels across multiple warehouses under conditions of demand uncertainty, ensuring a high probability of meeting that demand. The JCC-IM problem is formally defined as:

$$\textbf{JCC-IM}: \underset{\mathbf{x} \in \mathbb{R}^n}{\text{minimize}} \quad \mathbf{c}^\mathsf{T}\mathbf{x} \tag{72}$$

$$\text{subject to} \quad \textbf{Prob}(A\mathbf{x} \ge \theta + \omega) \ge 1 - \delta \tag{73}$$

$$G\mathbf{x} \le \mathbf{h}, \ \mathbf{x}^{\min} \le \mathbf{x} \le \mathbf{x}^{\max}, \tag{74}$$

where $n$ denotes the number of warehouses located in distinct regions, the decision variable $\mathbf{x}$ represents the inventory order quantity to be determined in advance for $n$ warehouse, in order to satisfy future demand. The vector $\theta$ encapsulates the historical average demand, and the term $\omega \sim p(\cdot)$ models the stochastic deviations from this average, capturing the inherent uncertainty of demand. The matrix $A$ characterizes the interdependencies among different warehouses, which may arise

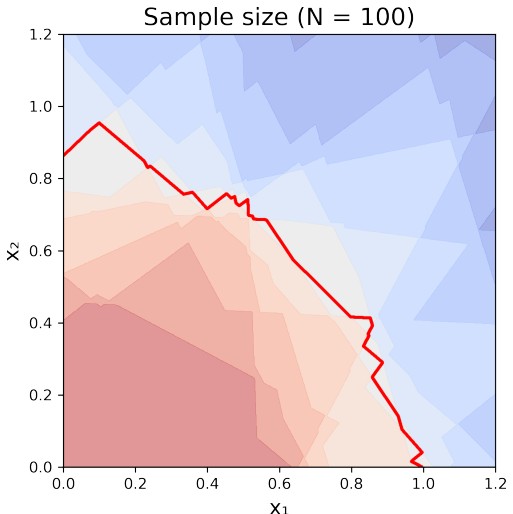 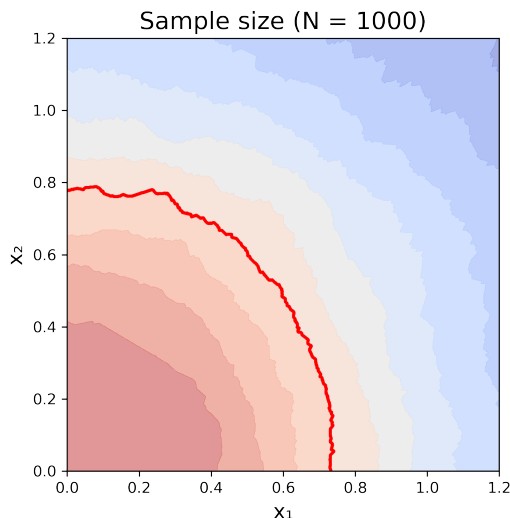

*Figure 10.* This figure visualizes a sample-based individual chance constraint defined as $\text{Prob}(\omega_1 x_1 + \omega_2 x_2 \leq 1) \geq 90\%$, where $\omega_1, \omega_2$ are independent Gaussian variables. The probability of satisfying this constraint is estimated using $N$ samples and the indicator function $I(\cdot)$ as: $\frac{1}{N}\sum_{i=1}^{N} I(\omega_1^i x_1 + \omega_2^i x_2 \leq 1) \geq 90\%$. The visualizations underscore the non-smooth geometry and optimization difficulty (PAS09). We remark that BP was tested in a high-dimensional scenario with 400 decision variables and joint constraints in our experiments.

from shared types of inventory or geographical proximity. The parameter $\delta$ specifies the acceptable risk level, thus ensuring that the probability of meeting demand across all warehouses is at least $1 - \delta$. The additional constraints, $G\mathbf{x} \leq \mathbf{h}$ and $\mathbf{x}^{\min} \leq \mathbf{x} \leq \mathbf{x}^{\max}$, represent warehouse-specific capacity limitations and inventory bounds, respectively.

The Joint chance constraint (JCC) represents the probability of the joint event of feasibility for each constraint as:

$$\mathbf{Prob}(A\mathbf{x} \geq \theta + \omega) = \mathbf{Prob}(A_1\mathbf{x} \geq \theta_1 + \omega_1, \cdots, A_m\mathbf{x} \geq \theta_m + \omega_m) \geq 1 - \delta \tag{75}$$

When $m = 1$ and with Gaussian uncertainty, the probability constraint can be reformulated into a second-order cone constraint (BBV04). However, in the general case, given the absence of an analytical reformulation for the JCC, we employ a Sample-Average (SA) approach to approximate the chance-constrained problem. This technique involves generating a finite set of scenarios $\{\tilde{\omega}_j\}_{j=1}^{N}$ from the underlying distribution of $\theta$. The SA variant of the JCC-IM is formulated as:

$$\mathbf{SA\text{-}JCC\text{-}IM} : \underset{\mathbf{x} \in \mathbb{R}^n}{\text{minimize}} \quad \mathbf{c}^\mathsf{T}\mathbf{x} \tag{76}$$

$$\text{subject to} \quad P_N = \frac{1}{N}\sum_{j=1}^{N} I(A\mathbf{x} \geq \theta + \tilde{\omega}_j) \geq 1 - \delta \tag{77}$$

$$G\mathbf{x} \leq \mathbf{h}, \ \mathbf{x}^{\min} \leq \mathbf{x} \leq \mathbf{x}^{\max} \tag{78}$$

where $I(\cdot)$ is the indicator function. A solution is deemed to have a probabilistic JCC feasibility guarantee if $P_N \geq 1 - \delta$. This empirical evaluation provides a practical measure of the reliability of the SA-based solution in adhering to the demand satisfaction requirements stipulated by the JCC-IM problem.

In practice, the problem can be solved as mixed-integer programming but is intractable for high-dimension problems with a large number of scenarios; therefore, for iterative-solver-based baselines, we solve the robust version of this problem by setting $\delta = 0$, such that the problem becomes convex with a large number ($N$) of constraints. But for BP methods, we can still project the solution to the chance constraint in (77) through bisection with easy feasibility checking shown in Alg. 1.

### D.1.3. ALTERNATING CURRENT OPTIMAL POWER FLOW (AC-OPF)

The Alternating Current Optimal Power Flow (AC-OPF) problem is pivotal in ensuring the efficient and safe operation of power grids. It requires real-time decision-making and adherence to operational constraints to maintain system integrity. The

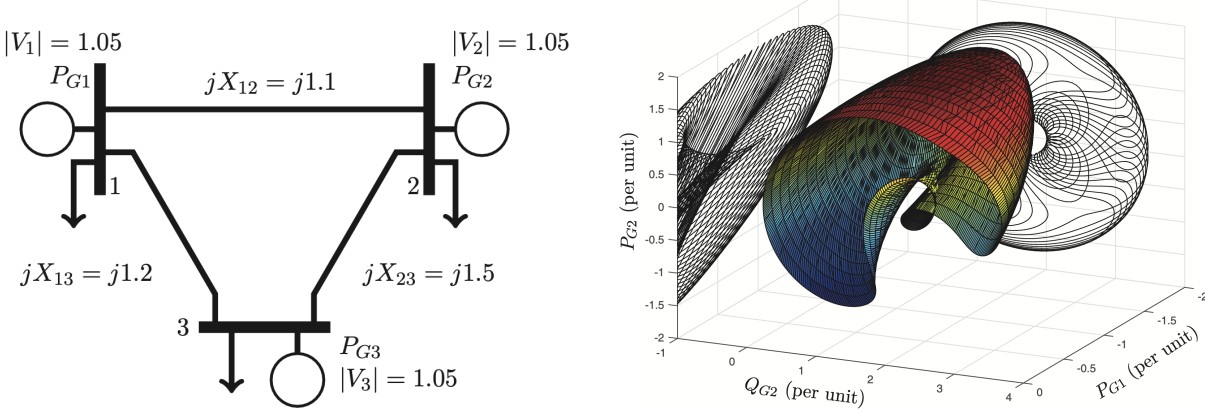

*Figure 11.* The **left** figure displays a simple 3-node power network, while the **right** figure illustrates a part of its constraint set (MH+19). These visuals highlight the complex geometry and inherent challenges of the ACOPF problem. It is noteworthy that in our experiments, BP was tested on a 200-node power network involving more than 1,000 constraints.

AC-OPF is inherently a non-convex Quadratically Constrained Quadratic Program (QCQP) and is recognized as NP-hard, posing significant computational challenges. The formal mathematical formulation of the AC-OPF problem is as follows:

$$\textbf{AC-OPF} : \min_{p_g, q_g, v} \quad p_g^\mathsf{T} Q p_g + b^\mathsf{T} p_g \tag{79}$$

$$\text{subject to} \quad |v_i(\bar{v}_i - \bar{v}_j)\bar{w}_{ij}| \le S_{ij}^{\max}, \quad \forall (i,j) \in \mathcal{E}, \tag{80}$$

$$(p_g - p_d) + (q_g - q_d)\,i = \operatorname{diag}(v)\bar{W}\bar{v}, \quad \forall i \in \mathcal{N}, \tag{81}$$

$$p_g^{\min} \le p_g \le p_g^{\max}, \; q_g^{\min} \le q_g \le q_g^{\max}, \; v^{\min} \le |v| \le v^{\max}. \tag{82}$$

where the power network comprises $n$ nodes, indexed by the set $\mathcal{N}$. The vectors $p_d, q_d \in \mathbb{R}^n$ represent the real and reactive power demand at each node, respectively. The vectors $p_g, q_g \in \mathbb{R}^n$ denote the real and reactive power generation, which are the decision variables of the optimization problem. The vector $v \in \mathbb{C}^n$ signifies the nodal voltage phasors. The admittance matrix $W \in \mathbb{C}^{n \times n}$ characterizes the physical properties and topology of the power network, with $\bar{W}$ denoting its complex conjugate transpose. The generation cost is represented by a quadratic function with matrix $Q \in \mathbb{R}^{n \times n}$ and vector $b \in \mathbb{R}^n$. The constraints include generation limits ($p_g^{\min}, p_g^{\max}, q_g^{\min}, q_g^{\max}$), voltage magnitude bounds ($v^{\min}, v^{\max}$), thermal line limits ($S_{ij}^{\max}$), and power flow balance equations. The set $\mathcal{E}$ denotes the set of edges (transmission lines) connecting the nodes in the power network. The equality constraint represents the complex power flow balance at each node, ensuring that the generation and demand are matched while accounting for power losses.

### D.2. Experiment Setting

**Computational Infrastructure**: All NN-based methods are implemented in Pytorch and executed on an Ubuntu server with an NVIDIA A800 GPU. Iterative algorithms were executed in parallel on an AMD EPYC 7763 64-Core Processor. For convex optimization problems, we employed the MOSEK optimizer under an academic license. The Joint Chance-Constrained Inventory Management (JCC-IM) problem was approximated using sampled scenarios and solved with MOSEK. Alternating Current Optimal Power Flow (AC-OPF) problems were addressed using the open-source PyPower toolkit (ZMS11). Additional experimental configurations are detailed in the respective sections and table footnotes.

**Baseline Methods**: We compare our approach against the following baselines:

- **Optimizer**: For convex optimization problems, we employ MOSEK as the baseline solver. For AC-OPF problems, we use PyPower (ZMS11) as the specialized solver.

- **NN**: A vanilla neural network that directly maps input parameters to solutions without feasibility guarantees. It is trained with an optimal solution dataset in a supervised setting.

- **Proj**: Infeasible predictions from NN are corrected via orthogonal projections. The projection problem is formulated and solved by Optimizer.

- **WS**: Infeasible NN predictions serve as warm-start initializations for iterative solvers, which may accelerate the convergence.

- **D-Proj**: The differentiable projection method from DC3 (DRK20), which employs gradient descent to minimize the constraint violation for constraint satisfaction. The gradient is derived via the automatic differentiation mechanism in PyTorch.

- **H-Proj**: Homeomorphic projection applied to infeasible predictions (LCL23). The invertible neural networks (INN) are trained in advance for each constraint type.

- **B-Proj**: Our proposed bisection-based projection (Algorithm 1) applied to predicted interior points for feasibility recovery.

**Evaluation Metrics**: We evaluate all methods using the following criteria on 1,024 test instances:

- **Feasibility**: The percentage of solutions satisfying all equality and inequality constraints within a tolerance of $10^{-5}$.

- **Optimality**: The relative solution and objective optimality gap, defined as $\frac{\|x-x^*\|}{\|x^*\|}$ and $\frac{|f(x)-f(x^*)|}{|f(x^*)|}$, respectively, where $f(x)$ is the objective value of the predicted/projected solution and $f(x^*)$ is the optimal objective value.

- **Runtime**: Wall-clock time for inference, including raw NN prediction and any post-processing steps.

### D.3. Hyper-parameters of NN and IPNN

We employ a fully connected neural network with residual connections (HZRS16; LLC24) and equality reconstruction (PZC19; DRK20), denoted as $F$, to predict optimal solutions or interior points for constrained optimization problems given input parameters $\theta \in \Theta$. The network architecture incorporates skip connections to facilitate gradient flow and a reconstruction module to enforce equality constraints.

**Data Generation for NN predictor**: Training and test datasets are generated by solving optimization instances across diverse parameter configurations using established solvers:

- **Convex problems**: MOSEK optimizer

- **AC-OPF problems**: PyPower (ZMSG97)

- **JCC-IM problems**: Sample Average Approximation (SAA) solved with MOSEK

**Loss Function for NN predictor**: The neural network is trained via supervised learning with a composite loss function that balances solution accuracy, constraint satisfaction, and objective quality:

$$\mathcal{L}(F) = \mathbb{E}_{\theta \sim \mathcal{D}} \left[ \|F(\theta) - x_\theta^*\|_2^2 + \lambda_1 \sum_j \text{ReLU}(g_j(F(\theta), \theta)) + \lambda_2 f(F(\theta), \theta) \right] \tag{83}$$

where:

- The first term minimizes prediction error with respect to optimal solutions

- The second term penalizes inequality constraint violations, with $g_j$ representing the $j$-th inequality constraint.

- The third term encourages objective function minimization to reduce the objective optimality gap directly.

- $\lambda_1, \lambda_2 > 0$ are hyperparameters controlling the trade-off between feasibility and optimality

**Constraint Handling in NN and IPNN:**

- **Equality constraints**: we employ variable selection and completion (detailed in Appendix A.4), which guarantees exact satisfaction for equality constraints.

- **Penalty function**: For standard differentiable inequality constraints $g(x) \leq 0$, we compute the constraint violation directly as $\|\mathrm{ReLU}(g(x))\|$, which naturally naturally vanishes when constraints are satisfied.

- **Positive semidefinite constraints**: For matrix variables $X \succeq 0$, We offer two penalty computation approaches
  - We can compute the exact penalty using negative eigenvalues via $\mathrm{ReLU}(-\mathrm{eig}(X))$, where eigenvalues are obtained through `torch.linalg.eigvalsh`. While this method provides exact gradients through automatic differentiation, it may encounter numerical instability (e.g., singularity) during training.
  - We can estimate the penalty as $\mathrm{ReLU}(-\sum_i v_i^\top X v_i)$ using linear measurements $\{v_i\}_{i=1}^k$, where the vectors $v_i$ are computed iteratively following (NSW22). This approximation offers improved computational efficiency and numerical stability compared to the eigenvalue approach.

**NN and IPNN Training**

We train the NN predictor following a regular ML training scheme with parameters in Table 4. The Interior Point Neural Network (IPNN) shares the same base architecture as the standard NN predictor, with the same input and output dimensions. Furthermore, for the IPNN training for constrained optimization problems, we directly initialize it with parameters from a trained NN predictor to reduce its training time. Detailed architectural specifications and hyperparameters for both models are provided in Table 4.

*Table 4.* Structure of IPNN/NN predictor in experiments

| Parameter | Value |
|---|---|
| NN/IPNN structure | |
| dimension of input layer | $d$ |
| dimension of output layer | $n$ |
| dimension of hidden layer | $\lfloor (d+n)/2 \rfloor$ |
| activation function | $\mathrm{ReLU}(\cdot)$ |
| number of layer | 3 |
| last-layer activation | $\mathrm{Sigmoid}(\cdot)$ |
| NN/IPNN training parameters | |
| number of training samples | 10,000 |
| number of testing samples | 1,024 |
| number of iteration | 10,000 |
| optimizer | AdamW |
| learning rate | 0.0001 |
| batch size | 64 |
| the coefficient for objective value | 0.001 |
| the coefficient for inequality penalty | 0.01 |
| the coefficient for robust margin | 0.01 |

# E. Supplementary Experiment Results

### E.1. IPNN Training and Bisection Projection over Various Constraint Sets

We evaluate the effectiveness of our BP framework on two challenging non-convex constraint sets.

**Test Cases**: We consider two geometrically distinct non-convex sets:

$$\text{Case 1:} \quad \mathcal{C}_1(\theta) = \{x \in \mathbb{R}^d \mid x^\top Q_i x + q_i^\top x + b_i \leq 0, \ i = 1, \ldots, 6\} \tag{84}$$

$$\text{Case 2:} \quad \mathcal{C}_2(\theta) = \bigcup_{i=1}^4 \mathcal{B}(c_i, r_i), \ \text{where} \ \mathcal{B}(c, r) = \{x \mid \|x - c\|_2 \leq r\} \tag{85}$$

where $\theta_1 = \{Q_i, q_i, b_i\}_{i=1}^6$ and $\theta_2 = \{c_i, r_i\}_{i=1}^4$ parameterize the constraint sets. Case 1 represents a ball-homeomorphic set defined by quadratic constraints, exhibiting complex non-convex geometry. Case 2 consists of a union of disjoint balls, presenting the additional challenge of disconnectivity—a property that violates the assumptions of many projection-based methods. For each case, we:

- Train the IPNN $\psi$ to learn the chebyshev centers.

- Evaluate bisection projection on unseen test parameters

- Compare against Homeomorphic Projection (H-Proj) using identical test instances

Results and Discussion. Results are presented in Figures 12 - 17. We make the following key observations:

- The robust margin maximization scheme provides two key benefits: it increases the interior-to-boundary distance, improving IPNN prediction feasibility on unseen samples, and enables the trained IPNN to predict central interior points for projections.

- For ball-homeomorphic constraint sets (Case 1), bisection projection achieves comparable performance to homeomorphic projection. However, when constraints are disconnected (i.e., non-ball-homeomorphic in Case 2), the homeomorphic projection fails to identify a valid INN for projection, resulting in infeasibility. In contrast, our bisection projection consistently identifies central interior points and maintains solution feasibility.

### E.2. Sensitivity Analysis on $\gamma$ for Out-of-Sample Feasibility and Projection Distance

We empirically evaluate the impact of the robust margin parameter $\gamma$ maximization on two critical performance metrics: (i) out-of-sample feasibility rates for interior point predictions on unseen parameter instances (Fig. 6), and (ii) incurred projection distances after bisection-based feasibility recovery (Fig. 7).

We compare three IPNN training strategies:

- **No $\gamma$**: Trained exclusively to minimize constraint violations without margin regularization ($\gamma = 0$).

- **Fixed $\gamma$**: Trained with a constant margin parameter (e.g., $\gamma = 10^{-2}$), analogous to randomized smoothing techniques employed in adversarial training.

- **Train $\gamma$**: Trained with our proposed $\gamma$ maximization regularization, initialized at $\gamma = 10^{-2}$.

To ensure statistical reliability, we trained each model configuration five times with different random seeds and report results with standard deviations.

### E.3. Sensitivity Analysis on Bisection Steps on Optimality Gap and Iteration Complexity

As established in Theorem 1, the bisection algorithm exhibits a linear convergence rate with low per-step computational cost. We empirically validate this theoretical property by systematically varying the number of bisection steps across multiple problem classes, as illustrated in Fig. 18.

The results confirm our theoretical analysis, demonstrating exponential reduction in optimality gap as the number of bisection steps increases, while computational time grows linearly. This favorable trade-off enables practitioners to select an appropriate number of iterations based on their specific accuracy requirements and computational constraints. Notably, most practical applications achieve acceptable convergence within 5-10 bisection steps, making our approach well-suited for real-time decision-making scenarios.

As shown in Fig. 2, when the line segment between an interior point and an infeasible prediction intersects the constraint boundary at multiple points, our bisection algorithm converges to one such intersection. The specific convergence point depends on the bisection parameter $\beta$, which controls the search granularity.

The bisection update rule can be generalized as:

$$\alpha_m = \beta \cdot \alpha_l + (1 - \beta) \cdot \alpha_u \tag{86}$$

where $\beta \in (0, 1)$ determines the bisection stepsize. The standard choice $\beta = 0.5$ yields binary search.

A smaller values (e.g., $\beta = 0.1$) create a conservative search biased toward the infeasible prediction, and the algorithm tends to converge to boundary points closer to the original NN prediction, better preserving the learned solution structure. However, this comes at a computational cost: The number of bisection steps increases under the same convergence tolerance.

Figure 19 empirically demonstrates this trade-off, showing that practitioners can tune $\beta$ based on their specific requirements for solution quality versus computational budget.

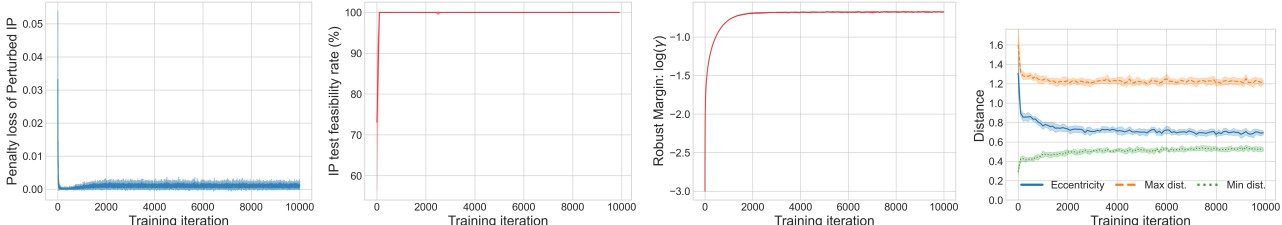

*Figure 12.* **Non-convex Case**: Perturbed penalty loss (Left) and IPNN prediction feasibility rate (Middle Left) during training. Robust margin $\log(\gamma)$ (Middle Right) and estimated eccentricity (Right) during training.

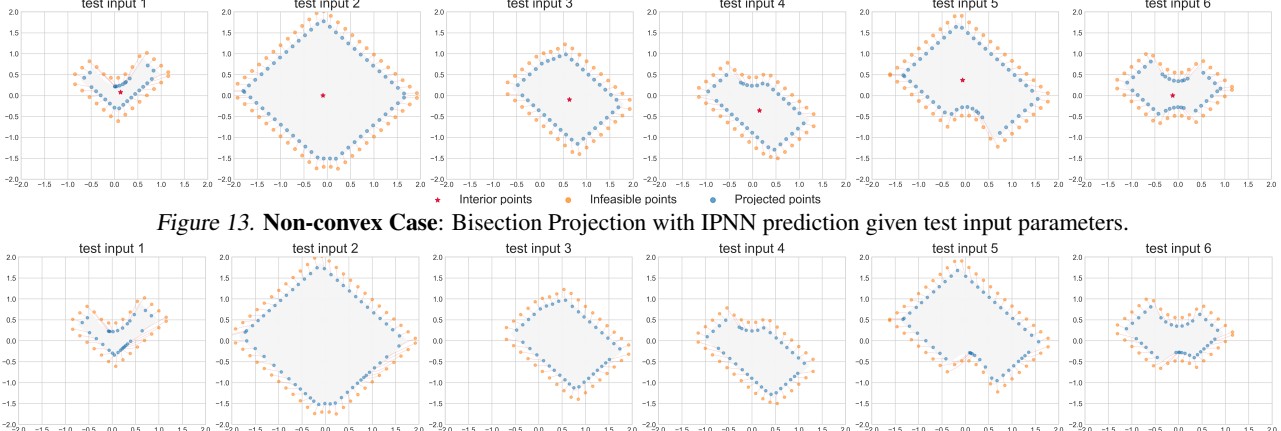

*Figure 13.* **Non-convex Case**: Bisection Projection with IPNN prediction given test input parameters.

*Figure 14.* **Non-convex Case**: Homeomorphic Projection with trained INN given test input parameters.

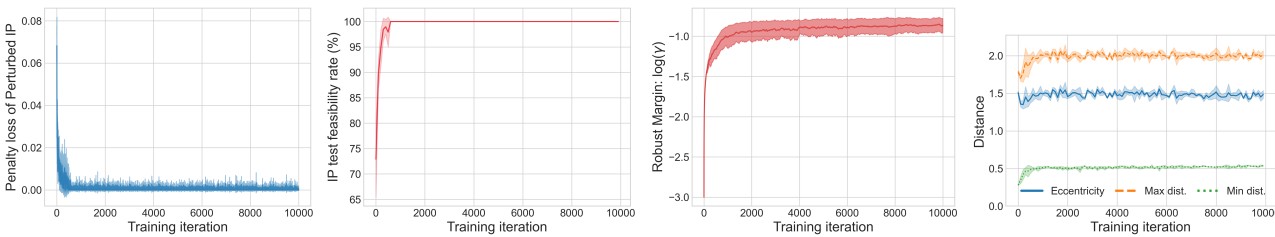

*Figure 15.* **Disconnected Case**: Perturbed penalty loss (Left) and IPNN prediction feasibility rate (Middle Left) during training. Robust margin $\log(\gamma)$ (Middle Right) and estimated eccentricity (Right) during training.

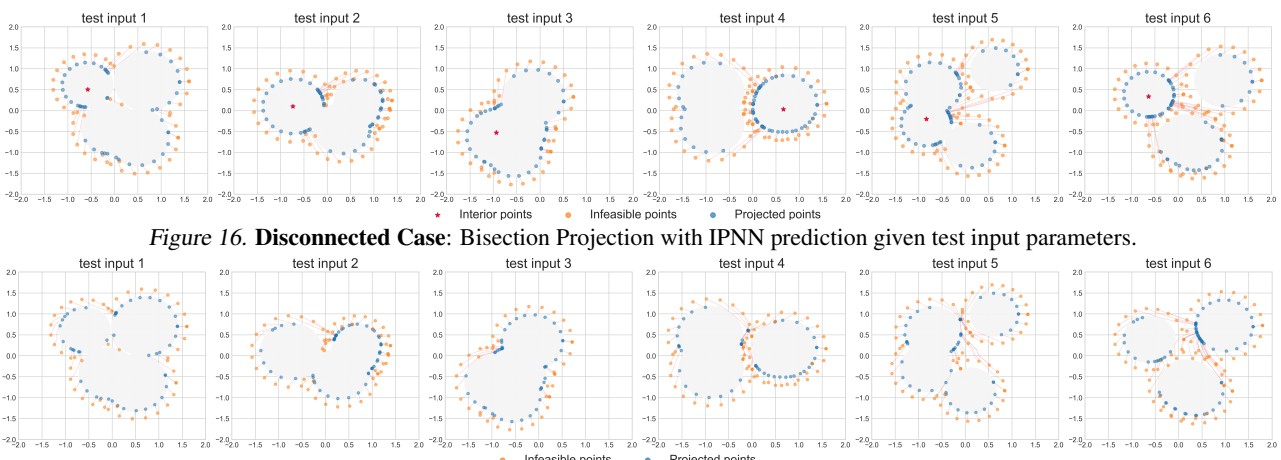

*Figure 16.* **Disconnected Case**: Bisection Projection with IPNN prediction given test input parameters.

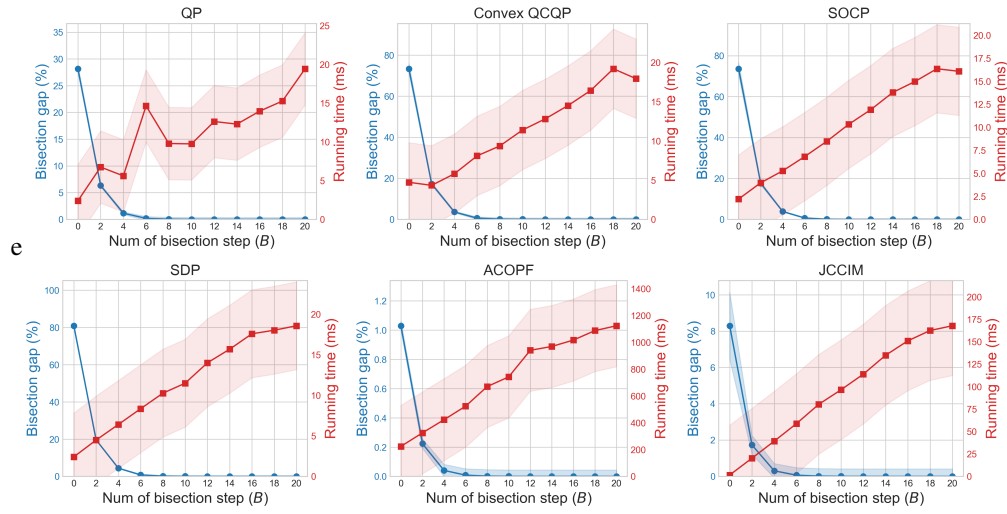

*Figure 17.* **Disconnected Case**: Homeomorphic Projection with trained INN given test input parameters.

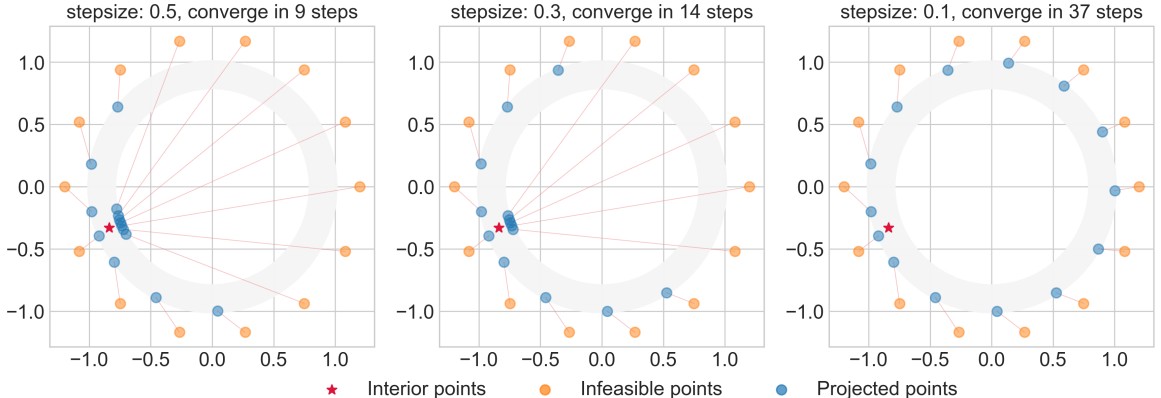

*Figure 18.* Effect of increasing bisection steps on optimality gap (top) and computational time (bottom) across four constraint classes. Results are averaged over initially infeasible neural network predictions. The panels show QP, QCQP, SOCP, SDP, AC-OPF, and JCC-IM problems sequentially, demonstrating consistent exponential convergence behavior with linear time complexity growth.

*Figure 19.* Effect of bisection parameter $\beta$ on (a) optimality gap and (b) iteration count across different problem instances. Smaller $\beta$ values achieve better solution quality at the expense of increased iterations.

