# OpenReview forum: "Efficient Bisection Projection to Ensure Neural-Network Solution Feasibility for Optimization over General Set"
_ICML.cc/2025/Conference — ICML 2025 poster_

### Official Review · Reviewer_8qqe · 2025-03-09

**Overall Recommendation:** 2

**Summary:**

The paper introduces a bisection procedure to achieve feasibility of solution outputs from neural networks applied to constrained optimization problems.

## update after rebuttal:
I updated my recommendation following the discussion period.

**Claims And Evidence:**

The paper provides a bisection procedure, establishes its claims, and presents numerical results to support its contribution.

**Essential References Not Discussed:**

no

**Experimental Designs Or Analyses:**

I only checked the theoretical analysis and methodology.

**Methods And Evaluation Criteria:**

The method comprises two elements: a bisection procedure and an IPNN training.
The bisection procedure is very simple and relies on basic ideas -- between a feasible point in the interior and an infeasible point there is a line in which there are points that are feasible so searching on that line can yield a closer feasible point.
IPNN is a neural network to predict interior points by minimizing some loss function with uniform sampling and Adam. Overall this is a heuristic method.

The evaluation measure is the distance between the optimal solution of the problem and the feasible point obtained from the infeasible solution output of the NN. This measure is not sufficiently motivated considering that global optimal solution of a nonconvex problem is unattainable and it requires some interior point reference.
Additionally, the bound established for this measure is not meaningful and seems quite straightforward.

**Other Comments Or Suggestions:**

no

**Other Strengths And Weaknesses:**

The paper proposes a technical procedure to tackle feasibility issues arsing the usage of NNs to solve constrained optimization problems.
It uses straightforward technical solutions with simple theoretical analysis.

**Questions For Authors:**

no

**Relation To Broader Scientific Literature:**

The key contributions are related to implementation and utilization of NNs as optimization tools for constrained optimization, making them, in my opinion, less relevant to research related literature.

**Theoretical Claims:**

The theoretical claims mainly follow from simple arguments and the underlying assumptions.
I briefly verified their correctness.

---

> ### Author Rebuttal · Authors · 2025-04-01
>
> **Response**:
>
> Thank you for reviewing our paper. We appreciate your feedback and the chance to address your concerns.
>
> We believe our application of the bisection procedure to neural networks, while simple, offers a practical solution to feasibility challenges in constrained optimization. The simplicity is intentional and enables broader implementation.
>
> Regarding your concerns about our evaluation metrics, we believe there may be a misunderstanding about our methodology. Our approach does not require global optimal solutions of nonconvex problems but rather focuses on improving feasibility while maintaining initial NN solution quality — a practical consideration for real-world applications of neural networks in optimization.
>
> Regarding our work's relevance within the research landscape, our method contributes to the advancement of machine learning approaches for solution generation in time-sensitive applications, a value that other reviewers have also acknowledged.
>
> In our detailed response below, we address your specific points. We also clarify the novelty, significance, and theoretical foundations of our work. We welcome your additional feedback and are committed to addressing your further comments in the following discussion period and improving both this work and our future research.
>
> We genuinely appreciate the time you and other reviewers have invested in evaluating our submission.
>
>
>
>
>
> ---
> > `C1: The bisection procedure is very simple and relies on basic ideas. IPNN is an NN to predict IPs by minimizing some loss function with uniform sampling and Adam. Overall, this is a heuristic method.`
> ---
>
> Thank you for your assessment. While our approach uses fundamental principles, simplicity is advantageous when effectively solving complex problems. Our contributions are:
>
> - Addressing research gaps in identifying quality interior points (IP) for general constraint sets and developing real-time IP prediction capabilities
> - A novel loss function design specifically targeting low-eccentricity interior points to minimize projection-induced optimality gaps
> - Theoretical analysis providing optimality bounds (Prop. 4.1, Theorem 1) and feasibility conditions (Prop. 5.1), establishing our method's mathematical foundation beyond mere heuristics
>
>
>
> ---
> > `C2: The evaluation measure is not sufficiently motivated considering that the global optimal solution of a nonconvex problem is unattainable, and it requires some interior point reference.`
> ---
>
> We acknowledge that global optimal solutions for general nonconvex problems are typically not guaranteed. However, our approach does not require global optimal solutions of nonconvex problems, but rather focuses on improving feasibility while maintaining initial NN solution quality — a practical consideration for real-world applications of neural networks in optimization.
>
> Moreover, our experiments evaluate solution quality by comparing objective values against state-of-the-art iterative solvers, providing direct empirical validation.
>
>
>
> ---
> > `C3: Additionally, the bound established for this measure is not meaningful and seems quite straightforward.`
> ---
>
> The bounds provided in Theorem 1 offer practical guidance for minimizing the overall optimality gaps in our framework. Its value lies in its decomposition of the error sources:
> - Initial prediction errors, which are well addressed by existing NN-based approaches;
> - Projection errors, which are the primary focus of our work;
> - Finite step bisection errors, which converge exponentially, demonstrating the efficiency of our framework.
>
> This decomposition not only clarifies the theoretical foundations of our approach but also provides insights for algorithm design and optimization. By identifying the projection error as a critical component that previous works have overlooked, we establish a clear direction for improving the performance of our framework.
>
>
>
>
>
> ---
> > `C4: The key contributions ..., in my opinion, less relevant to research-related literature.`
> ---
>
> We respectfully disagree with this assessment.
> - Neural networks for accelerating optimization problems represent an active and growing research area with significant interest from both machine learning and optimization communities, as evidenced by recent publications in top-tier venues, including ICML, NeurIPS, ICLR, and JMLR (see detailed discussions in Related Work Section.)
> - Our work addresses a fundamental challenge in this domain - ensuring the feasibility of NN predictions for constrained optimization - which has been identified as a key limitation in prior work. By providing a solution with theoretical guarantees, our paper makes a meaningful contribution to this important research direction.

---

> > ### Comment · Reviewer_8qqe · 2025-04-03
> >
> > Thank you for the detail rebuttal.
> >
> > I agree that the paper might provide contribution of interest to the ML community.
> > However, IMHO this contribution is of a technical nature that is less suited for the ICML venue.
> > Therefore I keep my recommendation as is.

---

> > > ### Author Response · Authors · 2025-04-09
> > >
> > > Dear Reviewer 8qqe,
> > >
> > > Thank you for acknowledging our work's potential contribution to the ML community.
> > >
> > > We respect your assessment regarding the technical nature of our contribution.
> > > We also believe that ICML has historically welcomed a diverse range of papers spanning the theoretical-practical spectrum and many technical advances presented at ICML have later enabled applied breakthroughs.
> > >
> > > Regardless of the final decision, we appreciate your engagement with our work and will take your feedback into consideration in future works.
> > >
> > > Sincerely,
> > > Authors

---

### Official Review · Reviewer_h2io · 2025-03-13

**Overall Recommendation:** 4

**Summary:**

One of the main challenges of ML-based solution generation in constrained optimization is to ensure feasibility. This paper describes a method to produce feasible solutions that are close to ML-generated potentially-infeasible solutions for compact constraint sets with non-empty interiors, with a focus on efficiency. Like in typical ML-based solution generation, we assume that the constraint set is parameterized and we have training data over this parameter. The general idea is to learn an interior point with low eccentricity -- though in the implementation this is approximated with learning the Chebyshev center -- and use a standard bisection method between this interior point and the predicted solution to find a feasible solution close to the predicted one. This is supported by theoretical results that bound the projection distance and optimality gap based on the eccentricity of the interior point and other parameters. Computational results demonstrate that this method is fast relative to baselines with little loss to the optimality gap in most cases.


### Update after rebuttal

I maintain my review after rebuttals and discussions. I appreciate the authors making the improvements discussed in the rebuttals. I do not have further comments to add.

**Claims And Evidence:**

The main claim of the paper is that the bisection projection method performs well for purposes of projecting a solution to a given feasible set. This is well-supported with a theoretical base and strong computational experiments. According to the experiments, this method provides feasible solutions with objective value generally comparable to the one obtained by orthogonal projection, which is the feasible solution closest to the predicted one, except that it is orders of magnitude faster. This type of computational result is valuable to make ML-based solution generation more practical.

Despite the strong end-to-end results, in my opinion there is a gap in the experimental evidence: the paper proposes the prediction of an interior point to produce the projected point, but does not evaluate this predictor, only the end-to-end method. I am interested in seeing if this approach is able to predict the actual Chebyshev center, or produce points with low eccentricity. One could compute the Chebyshev centers of these sets and compare the results. While it is perfectly valid to argue that obtaining an accurate approximation to a minimum-eccentricity point is not too important if the method works well end-to-end, I would say that such experiments would help a reader validate the theoretical reasoning behind this algorithm and ensure that each step of the way is working well. That said, although this gap is a weakness of the paper, I would not say it is a major weakness in view of the end-to-end results.

**Essential References Not Discussed:**

I am not aware of other references that need to be included in the paper.

**Experimental Designs Or Analyses:**

The experimental setup and analyses are generally solid with some room for improvement; more details are in the other sections.

**Methods And Evaluation Criteria:**

The proposed method is clean and reasonable. While I would not consider the ideas in this paper to be very sophisticated, this relative simplicity is also what makes it appealing for practical use if one already has the framework for ML solution prediction. I am not well-versed in this line of work, but based on a quick literature search, this method appears to be novel. Furthermore, I believe that the impact of this method is fairly significant. Having a fast projection method like this one facilitates the use of ML-based solution generation methods in applications with strict latency requirements, particularly given how general this approach is (albeit limited to continuous optimization).

Aside from the experimental gap discussed above, the end-to-end experimental section is solid and comprehensive, though there is some room for improvement on the presentation of the results. The baselines are all appropriate, there is a good variety of applications, and the sensitivity analysis experiments provide additional insight. My concerns are generally minor and I put them in the Questions section below.

**Other Comments Or Suggestions:**

Typos:
* Line 241: Missing closing braces.
* Prop. 5.2: Repeated "to to".
* Equation (44) is missing a subscript on \theta.

Editorial issues:
* Could you please add an explicit definition of g, if I have not missed it?
* I assume that Objective Gap (%) is averaged over the testing instances. Could you please mention it explicitly?
* The abstract mentions ensuring "NN feasibility". This was confusing at first read, as you just want to ensure solution feasibility, independently of the NN. I would suggest removing the "NN" from this expression.

**Other Strengths And Weaknesses:**

All strengths and weaknesses are discussed in other sections.

**Questions For Authors:**

1. Please address the concerns that were discussed above.

2. I understand that speedup is relevant for comparing methods, but please include the actual compute time as well. This can be difficult to evaluate without absolute numbers for context.

3. Could you comment on empirical robustness on the quality of the objective? While I see that the average optimality gaps are good, I am interested in knowing how often this approach might produce solutions that end up with poor solution quality. Or does it always produce good solutions? If there is anything that you could add to the paper on this topic, that would be great.

4. Could you add some rough estimates on training times?

5. At the end of p.7, the paper mentions that GPU-based processing accelerates constraint checking. Could you please clarify in the paper whether you are using GPUs for constraint checking in your computational experiments, or only for the neural network training and inference?

6. Could you please add a clarification in Fig. 4 if the 0-th step is the midpoint or the interior point? If it is the interior point, it seems that the predicted AC-OPF interior point is not bad in terms of objective value; can you comment on why?

7. If I understand correctly, the bisection method could converge to a point that is far from the NN solution if there are multiple options. For example, in Figure 2, the resulting point could lie in the boundary of the hole if the bisection point happens to fall inside it. This is of course only relevant for non-convex sets, but did you observe such a scenario occurring the applications you investigated? Is there no mechanism to avoid it? I imagine this might not be a significant issue for practical applications, but I am curious whether it could be.

8. It was not clear to me if the prediction-aware variant of this method was actually evaluated. I believe that the main results are for the prediction-agnostic variant, correct? Have you performed any evalutions for the prediction-aware version? The idea of learning the two points jointly is seems promising and I am curious if it performs well.

**Relation To Broader Scientific Literature:**

While I am not fully familiar with the literature of projection methods in this context, this paper does a good job at summarizing work in this area. In particular, the paper puts this method into context of previous works in Proposition 5.2.

**Theoretical Claims:**

The theoretical results support the effectiveness of the method, particularly showing that you can control the quality of the projection if you use an interior point of low eccentricity. I read through the theoretical proofs in the appendix superficially and did not check them in detail.

---

> ### Author Rebuttal · Authors · 2025-04-01
>
> We appreciate the recognition of our method's practical value for ML-based solution generation in constrained optimization problems, its theoretical foundation, and strong computational results. Below, we address specific points raised in the review.
>
> ---
> > `C1: Ability to predict low-eccentricity points/Chebyshev centers`
> ---
>
> We've conducted additional experiments to verify that our IPNN indeed produces approximated Chebyshev centers with low eccentricity. As shown in **Fig. 1-2** ([Link](https://anonymous.4open.science/r/Rebuttal_ICML_25-82D4/Reviewer%20h2io.pdf)),
>   - During IPNN training: the penalty loss for constraint violation decreases and the robust margin $\gamma$ increases. Thus, the minimum point-to-boundary distance increases and the eccentricity decreases
>   - After training, given new inputs, IPNN generalizes well to unseen input parameters, consistently producing central interior points.
>
> For high-dimensional problems in our main experiments, directly measuring eccentricity is challenging due to difficulties in sampling boundary points, which is why we focused on end-to-end performance metrics.
>
>
> ---
> > `C2: Definition of g(·)`
> ---
>
>  $g(x,\theta)=[g_1(x,\theta),\cdots,g_{n_{\rm ineq}}(x,\theta)]$ refers to the constraint function for inequality constraints, as noted in Appendix A. We'll add this formal definition to the problem formulation section.
>
>
> ---
> > `C3: Objective Gap calculation.`
> ---
>
> Yes, the Objective Gap (\%) is averaged over all testing instances. We'll clarify this in the experimental section.
>
>
>
> ---
> > `C4: Solution feasibility beyond NN applications`
> ---
>
> While we focus on ensuring NN solution feasibility for constrained optimization, we acknowledge the broader applicability of our approach. We'll discuss generalizations to other problems in the revised manuscript.
>
>
>
> ---
> > `Q1: Actual baseline computation time and IPNN training time`
> ---
>
> We provide the computation time for baseline iterative solvers shown in **Table 1**, and the IPNN training time for several optimization problems in **Table 2** via [Link](https://anonymous.4open.science/r/Rebuttal_ICML_25-82D4/Reviewer%20h2io.pdf). The training time ranges from 30 seconds to 8 minutes.
>
>
>
>
>
> ---
> > `Q2:  empirical robustness and possible failure cases`
> ---
>
> Based on Prop. 4.1, our method can produce significant optimality gaps in three scenarios:
>
>   - (a) "Thin" constraint sets with unavoidably large eccentricity
>   - (b) Poorly selected interior points (e.g., near boundaries) causing large eccentricity
>   - (c) Large initial NN prediction errors
>
> We provide **Fig. 3** ([Link](https://anonymous.4open.science/r/Rebuttal_ICML_25-82D4/Reviewer%20h2io.pdf)) to illustrate these cases, which we'll include in our revised manuscript.
>
>
>
> ---
> > `Q3: GPU usage for constraint checking`
> ---
>
> Yes, we use GPU for all NN-based methods and related processing (e.g., constraint checking in B-Proj and gradient evaluation in D-Proj) for fair comparison.
>
>
>
> ---
> > `Q4:  0-th step in bisection and AC-OPF objective gap`
> ---
>
> The 0-th step returns the interior point.
>
> Regarding AC-OPF's objective gap: 2.5\% gap (at the 0-th step) is significant in power systems because:
>   - Its objective represents generation cost for active power (a subset of all decision variables)
>   - Other variables primarily maintain physical power balance constraints, not in the objective
>   - In the power systems community [1,2], such gaps are meaningful as base generation costs are typically high (e.g., $260,200 for a 793-node network [2])
>
> [1] P., X., ... Deepopf: A DNN approach for security-constrained dc optimal power flow. IEEE TPS, 36(3). 2020
>
> [2] B., S., ... The power grid library for benchmarking ACOPF algorithms. arXiv. 2019
>
>
>
> ---
> > `Q5: Bisection convergence with multiple intersections`
> ---
>
> The bisection algorithm can converge to one of multiple intersection points when they exist (Sec. 4.1). Although we didn't observe this in our non-convex engineering problems, we acknowledge this possibility.
>
> To prevent convergence to points far from the initial prediction, we can adjust the bisection step size:
> - Standard bisection uses step size 0.5: $a_m = (a_l+a_u)/2$
> - Using larger step sizes (e.g., 0.75): $a_m = 0.25a_l+0.75a_u$ creates a trajectory starting closer to the initial point.
> - This approach promotes convergence to boundary points nearer to the original prediction.
>
> **Fig. 4** ([Link](https://anonymous.4open.science/r/Rebuttal_ICML_25-82D4/Reviewer%20h2io.pdf)) demonstrates that adjusting the step size from 0.5 to 0.75 helps avoid convergence to distant points.
>
>
>
>
> ---
> > `Q6:  Prediction-aware variant evaluation`
> ---
>
> We only evaluated the prediction-agnostic version because it doesn't require optimal solution training data and already achieves small optimality gaps. Future work will explore prediction-aware variants for applications where objective values are highly sensitive to decision variables or where NN predictors struggle with initial approximations.

---

> > ### Comment · Reviewer_h2io · 2025-04-04
> >
> > Thanks for the rebuttal. I have read through the responses and they generally address my questions. I only have comments on a few of them:
> >
> > * **C1:** Thank you for running these experiments. I was somewhat hoping to see an evaluation over the benchmark set, but this is also fine. Please add this to the Appendix of the paper. (Note: Fix subfigure references in Fig. 1.)
> >
> > * **Q1:** Thank you for the table. Just to make sure I understand this: for example, the QP solver takes on average 62ms, and your method is 2193x faster, so it takes 0.028ms? These seem to be very low solve times so I am wondering if I am misinterpreting this. In any case, if you find space, I suggest adding the data from Table 1 to the main text (rather than Appendix), since I believe this is important contextual information. Table 2 is fine as part of the Appendix.
> >
> > * **Q2:** These figures are interesting to understand failure cases and I appreciate you adding them to the Appendix. I just wanted to comment that my original intent was understanding whether such failure cases occur with your method, i.e. how often the method produces low quality outliers. This is actually addressed by your Fig. 1 to reviewer cZEk. In particular, adding the margin seems to make your method much more robust, which is a positive sign.
> >
> > Please make sure to add all these details to the paper. I particularly liked seeing the results of the ablation study on $\gamma$ suggested by Reviewer cZEk. I maintain my assessment of accept.

---

> > > ### Author Response · Authors · 2025-04-06
> > >
> > > Dear Reviewer h2io,
> > >
> > > Thank you for your thoughtful feedback and for maintaining your acceptance recommendation.
> > > We appreciate your careful review and have addressed your comments as follows:
> > >
> > > - Response to **C1**:  We appreciate your interest in our additional experiments. We will incorporate these results in the Appendix and fix the subfigure references.
> > > - Response to **Q1**: Your interpretation of our results is correct.
> > >     - The QP solver (MOSEK) takes **63s in total** with full parallelization and 62ms per instance (divided by the number of instances), while our method takes approximately **29 ms in total** (0.028ms per instance).
> > >     - These remarkably low solve times result from efficient GPU batching processes, highlighting a key advantage of neural network-based approaches [1].
> > >
> > >     We agree this is important contextual information and will move the actual **total/average solving times** to the performance table in the main text of the revised manuscript.
> > > - Response to **Q2**:  We're pleased you found the failure case analysis helpful. Your observation about Fig. 1 demonstrating improved robustness with margin inclusion is indeed correct. This confirms that our method effectively reduces low-quality outliers, addressing the core concern in your original question.
> > >
> > > We will incorporate all these details and additional experimental results in the revised manuscript, including the ablation study on $\gamma$ suggested by Reviewer cZEk.
> > >
> > > Thank you again for your constructive feedback throughout the review process.
> > >
> > > Sincerely,
> > > Authors
> > >
> > > [1] Donti, P. L., Rolnick, D., & Kolter, J. Z. DC3: A learning method for optimization with hard constraints. ICLR 2021.

---

### Official Review · Reviewer_cZEk · 2025-03-18

**Overall Recommendation:** 4

**Summary:**

The paper proposes a technique that enables projecting the outputs of neural nets onto arbitrary compact sets. The goal is for the neural net outputs to be feasible with respect to constraints that are often found in convex and non-convex optimization settings. The approach adopted relies on an efficient bisection projection.

The key idea is that if we want to project to a specific set, if we have access to an interior point of that set, given a neural net prediction outside that set, we can consider the line segment from the interior point to the prediction and move along this direction until the boundary is intersected. In order to improve the efficiency of the method, the paper aims to use interior points that have low eccentricity, which basically implies that the point is more 'central' in the feasible set.

Finding such points is generally a hard problem and the authors resort to a relaxation that uses the Chebyshev center which finds a points that maximize the minimum distance from the boundary. To further improve efficiency, they train a neural network for this task using a suitable loss and provide sufficient conditions for this training to yield a NN that produces feasible interior points.

The method is evaluated on various convex and nonconvex problems and it consistently yields the fastest results with guaranteed feasibility and competitive objective values.

**Claims And Evidence:**

The claims are generally well supported and the paper provides clear explanations and proofs for its claims.

- My main issue has to do with the use of eccentricity. On a formal level, since there are no assumptions on the function $f$ other than continuity, does minimizing the projection distance matter? The function can be nonconvex (it doesn't even have to be smooth), so unless the neural net prediction is already extremely close to the optimal point, the value of the objective could change significantly after projecting. I would expect the projection distance minimization coupled with some kind of smoothness assumption on $f$ to be more meaningful, or have I misunderstood something?


- Conceptually, it would be nice to provide some examples of important problems where H-proj can't be applied while B-proj can. This will help emphasize how the generality of this work yields important benefits.

**Essential References Not Discussed:**

There is a recent paper that handles linear constraints that I believe is not mentioned in the related work. See also references in that paper and its table 1.

Zeng, Hongtai, et al. "GLinSAT: The General Linear Satisfiability Neural Network Layer By Accelerated Gradient Descent." arXiv preprint arXiv:2409.17500 (2024).

**Experimental Designs Or Analyses:**

- I think the use of eccentricity is not well supported by the experiments. In figure 3, we see that with or without it, a certain number of samples is necessary to achieve 100% feasibility. In those plots, at least in terms of feasibility, it doesn't seem to make a difference. I believe a more interesting experiment is an ablation on $\gamma$. Namely, what do the objective gaps look like when training the IP predictor without it? Is it necessary to deal with this issue or could any interior point work just as fine with this approach?
I can see the appeal around the eccentricity argument though I can't help but wonder how it's ever 'cashed out' experimentally. It would be good to see a more careful experimental evaluation of its contribution.

**Methods And Evaluation Criteria:**

The proposed evaluations and methods make sense for the problem and are in line with the literature.

**Other Comments Or Suggestions:**

n/a

**Other Strengths And Weaknesses:**

Overall, I think this approach balances efficiency and optimality well while improving significantly over previous work. At its core, the approach is fairly simple which makes it even more appealing. The paper is nicely written and it frames its contribution well in the context of related work.

On the other hand, to reiterate my main issue: I think the use of eccentricity (or a proxy for it anyway) should be better motivated and the experiments should also reflect its utility since a significant portion of the paper is spent discussing this. I can see at an intuitive level why you would like interior points with better 'centrality' and it helps minimize projection distance but, unless I misunderstood something, I don't think (as I explained above) a particularly compelling formal argument is made in the paper for its benefits.

In any case, I lean towards accepting this so I start with a tentative score and I will reconsider once the authors respond to the points/questions I brought up.

**Questions For Authors:**

- Can you explain how you encode the SDP feasibility for the NN that predicts interior points? How do you enforce the PSD constraint on the matrix variable, i.e., what does the loss function look like in that case.

**Relation To Broader Scientific Literature:**

This paper provides a quite fast and general method for ensuring the feasibility of neural net outputs. Based on the experimental results, this paper achieves the best tradeoff between speed and optimality of solutions.

**Theoretical Claims:**

I have checked the claims and have briefly gone over the proofs. They seem correct.

---

> ### Author Rebuttal · Authors · 2025-04-01
>
> We appreciate the recognition that our approach balances efficiency and optimality while improving over previous work, and that the simplicity of our method adds to its appeal. Below, we address each specific point raised in the review.
>
> ---
> > `C1:  Assumptions on the objective function and optimality gap`
> ---
>
> The reviewer raises an excellent point about the relationship between solution distance and objective gap.
>
>   - We use solution distance as our primary metric for theoretical analysis, where the optimality gap is bounded by initial prediction error and projection distance. For **Lipschitz** objective functions, solution distance directly translates to bounds on objective value gaps, as demonstrated in our experiments with linear/quadratic objectives.
>
> We acknowledge that for general non-smooth objectives, small solution distances might still yield large objective gaps. In our revision, we will clarify this relationship and specify the applicable scenarios of objective functions.
>
>
>
>
> ---
> > `C2:  examples where H-proj can't be applied while B-proj can.`
> ---
>
> The H-Proj method [1] is limited by its ball-homeomorphic assumption. Our B-proj method can handle several practically relevant constraints that H-proj cannot:
>   - Non-simply-connected feasible regions (with "holes" or disconnected components) occurring in AC Optimal Power Flow problems [2] and rocket landing problems with exclusion zones [3].
>   - We provide a visual example in **Fig. 2** ([Link](https://anonymous.4open.science/r/Rebuttal_ICML_25-82D4/Reviewer%20cZEk.pdf)) showing where H-proj incurs significant optimality loss while our method succeeds on such topologically complex regions
>
> [1] L., C. L. Homeomorphic Projection to Ensure Neural-Network Solution Feasibility for Constrained Optimization. Journal of Machine Learning Research, 2024.
>
> [2] M. H... A survey of relaxations and approximations of the power flow equations. Foundations and Trends® in Electric Energy Systems, 2019
>
> [3] A. C. B.  Lossless convexification of nonconvex control bound and pointing constraints of the soft landing optimal control problem. IEEE TCST, 2013
>
>
> ---
> > `C3:  ablation stufy of $\gamma$ on optimality gap`
> ---
>
> We appreciate this valuable feedback. We would like to clarify that:
>
>   - Fig 3 validates Prop. 5.1, showing that robust margin $\gamma$ reduces the number of training samples needed for feasibility under unseen inputs. With the same training sample, IPNN trained with $\gamma$ consistently achieves better out-of-sample feasibility rates.
>   - We agree that directly evaluating objective gaps would better demonstrate our eccentricity-minimizing targets' effectiveness. We've provided additional results in **Figure 1** ([Link](https://anonymous.4open.science/r/Rebuttal_ICML_25-82D4/Reviewer%20cZEk.pdf)) showing how eccentricity minimization specifically improves solution (worst-case gap: 12\% (no $\gamma$) vs 5\% (with $\gamma$) in QP instance).
>
>
> ---
> > `C4:  recent paper on linear constraints (GLinSAT)`
> ---
>
> It presents an efficient differentiable projection layer for general linear constraints. We will cite this work, along with the related works from their Table 1, and discuss their relevance in our revised manuscript.
>
>
>
>
> ---
> > `C5:  eccentricity should be better motivated. formal argument for benefits  eccentricity`
> ---
>
> Thanks for your feedback on the formal argument of the benefit of eccentricity minimization.
>
> Our theoretical analysis formally establishes the benefits of eccentricity minimization:
>
> - Prop. 4.1 shows the relationship between eccentricity and projection-induced distance, motivating our search for low-eccentricity interior points.
>  - Prop. 5.2 demonstrates how robust margin reduces training samples needed to achieve feasibility guarantees on unseen inputs
>
> We'll further clarify the motivations of eccentricity with illustrations (**Fig. 3** ([Link](https://anonymous.4open.science/r/Rebuttal_ICML_25-82D4/Reviewer%20cZEk.pdf)) and explicitly present formal arguments for eccentricity minimization in bullet points in revised Section 4.2, reinforcing these points in our analysis.
>
>
>
>
>
> ---
> > `Q1:  SDP feasibility and how to enforce the PSD constraint`
> ---
>
> For SDP problems (linear equality and PSD inequality):
>   - IPNN predicts independent variables and reconstructs dependent ones using equality constraints (Appendix A), then reshapes the vector prediction into matrix form.
>   - For inequality constraints, we use penalty methods to minimize constraint violation.
>
> For PSD constraint:
>   - We use the matrix-sketching [1] to capture constraint violations by $v^TXv$ in a differentiable way, where $v$ is iteratively calculated.
>   - By maximizing $v^TXv$ when negative, IPNN learns to find interior points for PSD constraints.
>   - During bisection projection, we directly check PSD feasibility using torch.linalg.cholesky_ex()
>
> [1] N., D., S., W., W., D. P. Testing positive semidefiniteness using linear measurements. IEEE FOCS. 2022

---

> > ### Comment · Reviewer_cZEk · 2025-04-07
> >
> > OK nice, please make sure you specify those details about the PSD constraint in the paper. In fact, I think highlighting the need for (essentially) a separation oracle and discussing the potential cost of that is essential in this approach.  For example it is clear that you tested on small SDPs and my guess is that it has to do with the problem of checking feasibility fast for large instances there. Your assumptions rule out constraints that show up in discrete optimization problems (and you do mention that plus the concern about the cost of evaluating the feasibility of a point) but I think this is an important limitation of the method that should be given room for discussion in order to also encourage future work.
> >
> > On the other hand, I can imagine that modifications of this method could potentially provide a possible direction for the discrete case too and the idea considerably simplifies previous projection methods which I find quite appealing. In my view this is a nice paper with a simple idea that works quite well so I recommend acceptance

---

> > > ### Author Response · Authors · 2025-04-08
> > >
> > > Dear Reviewer cZEk,
> > >
> > > Thank you for your thoughtful feedback and recommendation.
> > >
> > > We appreciate your insights regarding the PSD constraint and separation oracle aspects.
> > >
> > > - We will ensure the final version includes all details about the PSD constraint implementation and the additional experiments presented in the rebuttal.
> > >
> > > - As suggested, we will incorporate a discussion highlighting the necessity of a separation oracle for feasibility checking and analyze its computational complexity across various constraint sets, with particular emphasis on SDP problems.
> > >
> > > - We acknowledge the limitations of current methods that focus primarily on continuous domains, and will discuss the potential of our framework for discrete/combinatorial optimization as a significant future direction (e.g., discussing how applying SDP can serve as a continuous relaxation for combinatorial problems).
> > >
> > >
> > > Thank you again for your constructive feedback throughout the review process.
> > >
> > >
> > > Sincerely, Authors

---

### Decision · Program_Chairs · 2025-05-01

**Decision:**

Accept (poster)

**Comment:**

This paper tackles an important problem when using neural networks for optimization tasks: how to make sure that the output is a feasible solution? The authors provide an algorithm to practically overcome this issue and conduct a theoretical analysis and a computational study. The significance of this work was judged differently by the different reviewers, with the main criticism by one reviewer being that the method is simple and the theoretical analysis straightforward. The discussion was productive though, with reviewers, while not agreeing, being active and understanding for the other perspective. In the end, I follow the argumentation of the two reviewers recommending accept, saying that a simple but effective method is indeed very useful and a theoretical analysis, even if not too technical, definitely appreciated. This work is certainly a solid approach to ensure feasibility, the paper is well-written and meets the quality requirements for ICML.